ecology/plant science

traditional resource management, geophyte, Pacific Northwest, harvesting, foragers

**Authors for correspondence:**
Molly Carney
e-mail: molly.carney@wsu.edu
Jade d'Alpoim Guedes
e-mail: jguedes@ucsd.edu

# Harvesting strategies as evidence for 4000 years of camas (*Camassia quamash*) management in the North American Columbia Plateau

Molly Carney[1], Shannon Tushingham[1], Tara McLaughlin[2] and Jade d'Alpoim Guedes[3]

[1]Department of Anthropology, Washington State University, College Hall, Pullman, WA 99164, USA
[2]Department of Natural Resources, Kalispel Tribe of Indians, PO Box 39, Usk, WA 99180, USA
[3]Department of Anthropology, Scripps Institution of Oceanography, University of California San Diego, 9500 Gillman Drive, La Jolla, CA 92093, USA

(iD) MC, 0000-0003-1535-7363

One of the greatest archaeological enigmas is in understanding the role of decision-making, intentionality and interventions in plant life cycles by foraging peoples in transitions to and from low-level food production practices. We bring together archaeological, palaeoclimatological and botanical data to explore relationships over the past 4000 years between people and camas (*Camassia quamash*), a perennial geophyte with an edible bulb common across the North American Pacific Northwest. In this region throughout the late Holocene, people began experimenting with selective harvesting practices through targeting sexually mature bulbs by 3500 cal BP, with bulb harvesting practices akin to ethnographic descriptions firmly established by 1000 cal BP. While we find no evidence that such interventions lead to a selection for larger bulbs or a reduction in time to maturity, archaeological bulbs do exhibit several other domestication syndrome traits. This establishes considerable continuity to human intervention into camas life cycles, but these dynamic relationships did not result in unequivocal morphological indications of domestication. This approach to tracking forager plant management practices offers an alternative explanatory framework to conventional management studies, supplements oral histories of Indigenous traditional resource management and can be applied to other vegetatively propagated species.

# 1. Introduction

For almost all human groups through time and space, geophyte plants offer important and reliable sources of carbohydrates, nutrients and economic products. Geophytes, with their edible underground storage organs, are hypothesized to have been of critical importance to early hominin development [1] and use by *Homo sapiens* has been documented as early as 170 000 years ago [2]. Despite this, the complex geographical and historical pathways of managed and domesticated geophytes are still poorly understood [3]. Within North America, establishing timelines and trajectories of geophyte plant use and cultivation by hunter–gatherer groups has been particularly challenging [4,5], notwithstanding strong ethnohistoric evidence for their place as staple food sources in many Indigenous economies.

Understanding geophyte management practices in foraging societies and when these entanglements between climate, environment and food resources become deeply embedded in sociocultural systems is crucial to our collective knowledge of what happens when management does not lead to selection or domestication [6]. There are few studies which focus on these alternative human–plant relationships. We see several reasons for this. Most palaeoethnobotanical research examines human exploitation of plants, primarily through the lens of domestication and selection, rather than through reciprocal interactions between humans and plants [7–9]. The domestication process itself is viewed as the result of phenotypic and genetic changes in cultivars that are different from unmanaged populations [10], a process made even more difficult to study with the low chances of geophyte preservation in the archaeological record. It is now commonly accepted that these processes are coevolutionary, multi-directional pathways or entanglements between human and plant or animal species [11–13], and non-human species may also have a role in selection for domestication syndrome traits [14]. As the centres of domesticate origins increase in number and are continuously refined [10,15], it has become evident that varying ecologies, niches, aspects of plant physiology and cultural traditions have resulted in many different pathways to domestication and domestication syndromes. Indeed, the phenotypic and genetic changes we see today in contemporary domesticated plant species are probably the end results of various conscious or unconscious decisions and behaviours occurring over millennia [16].

In western North America, it is widely recognized that Indigenous hunting and gathering communities managed plants through a variety of practices and cultural institutions [17,18]. Signatures of anthropogenic burning [19], seasonal harvesting [20], archaeological garden features [21,22] and plant species range extensions [4,23–25] have been used to document past plant management practices in western North America. And while ethnographic work and oral histories attest to the presence of intimate human–plant symbiotic relationships for many plant food species, until now studies have not produced corresponding direct empirical evidence in the form of plant remains for significant archaeological geophyte management practices. Below we report archaeological evidence for substantial management of a cultural keystone geophyte species [26] and demonstrate considerable time depth of ethnohistoric plant stewardship practices that do not lead to domestication pathways.

*Camassia quamash* (Pursh) Greene, commonly known as camas, is a flowering herbaceous perennial with an edible bulb that is considered within many Northwest Indigenous communities to be among the most important plant food source [27]. Many have hypothesized that past Indigenous management practices positively impacted camas growth and production and that the plant may have been semidomesticated [28]. In fact, over 100 years ago renowned botanist Luther Burbank's experiments illustrated that traits within camas species (*Camassia* spp.) are easily selected for and produced notable increases in bulb sizes after only two generations [29]. Archaeological and genetic studies, however, have not yet produced evidence for human-mediated change within the plant's genotype or phenotype [30,31]. Here, we present archaeological evidence from the Pend Oreille Valley, in northeastern Washington State (figure 1), for two distinct camas harvesting patterns, one of which reflects conscious human intervention into the plant's life cycle. We show that harvesting strategies changed throughout our 4000-year study period, and further illustrate that such human–plant interactions resulted in a stable yet dynamic subsistence system in which people and camas coexisted in a symbiotic relationship that did not result in dependence on humans for reproduction or other phenotypic change.

# 2. Human relationships with camas life cycles

The North American genus *Camassia*, taxonomically classified as the Asparagaceae family, comprises five species with ranges across the United States and southern Canada [32,33]. *Camassia quamash* (common camas) is the most widespread species across the Pacific Northwest. Common camas is a bulbous

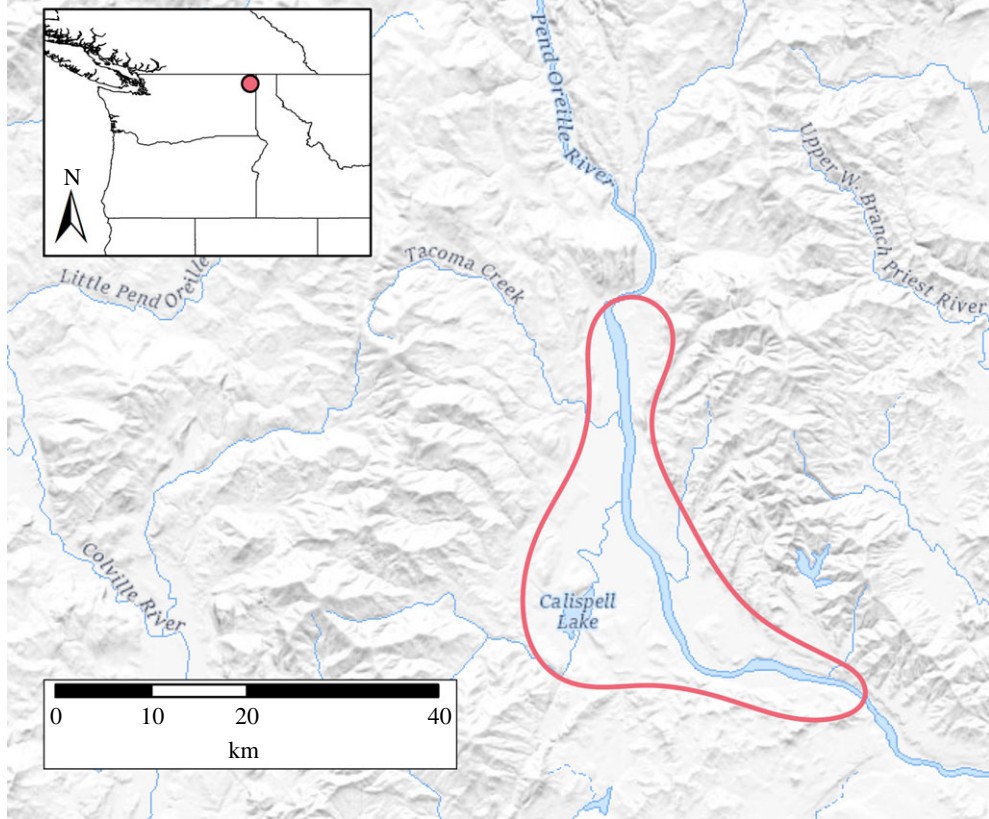

**Figure 1.** Relief map of the Pend Oreille Valley, northeastern Washington State, with location of valley outlined in light red.

perennial with inflorescences of large, blue flowers, which prefers to live in colonies in poorly drained fields or prairies with seasonal xeric moisture regimes (figure 2). The bulbs are perennating underground storage organs that consist of a shortened stem and leaf bases, as well as a basal plate from which fibrous roots emerge. These bulbs contain high amounts of inulin and are usually prepared by steaming or roasting in earth ovens for long periods of time (24–72 h) at high temperatures to convert the complex carbohydrates into easily digestible fructose [34]. Both substantial soil moisture from winter through spring and unrestricted drainage are necessary for optimal growth [35]. Excessive summer soil moisture or extreme heat will reduce bulb yields, while cold temperatures do not appear to impact productivity. Moreover, common camas needs 42–100 days of cold temperatures (less than 5°C) to germinate [36].

Ethnographic evidence indicates that camas was intensely managed and highly valued throughout the Northwest [25,27]. Contemporary practice and historic records indicate that camas was harvested with wooden digging sticks during the summer months after flowers had faded but stalks and seed capsules were still visible [27]. During bulb harvests, seed capsules and dried stalks were replanted along with any uprooted immature bulbs. Fields were also intentionally burned in some places. Such practices not only aerated soils but kept fields clear, removed competitors and enhanced plot productivity [28]. It is unknown when these practices became established, or how they may have changed throughout time. Previous research has established that camas has been consumed for over 8000 years [30], was traded [25] and had peaks and nadirs in consumption [37]. Camas production and storage is widely assumed to have been intensified by 4000–3000 years ago within the Columbia Plateau [37,38].

Several *Camassia* spp. life-history traits, including bulb anatomy and time to maturity, are important to understanding human management. Camas plants reproduce primarily by seed, but it is estimated that 15% of sexually mature bulbs will reproduce asexually through offset bulblets [28,37]. Seedlings take between 4 and 5 years to reach sexual maturity [36]. Bulbs have interior layered leaf scales and consist of two parts: a mother bulb and its enclosed daughter bulb. In late winter or early spring, new leaves are developed as part of the daughter bulb. These tissues are completely replaced annually [28]. While camas bulbs do not increase in size at a linear rate [28], leaf scales grow in regular layers and increase in number over time. Sexually mature bulbs of 4–5 years or greater in age will have 3–7 leaf scales while immature bulbs of 3 years or less often have 1–3 leaf scales [36,37] (figure 2c).

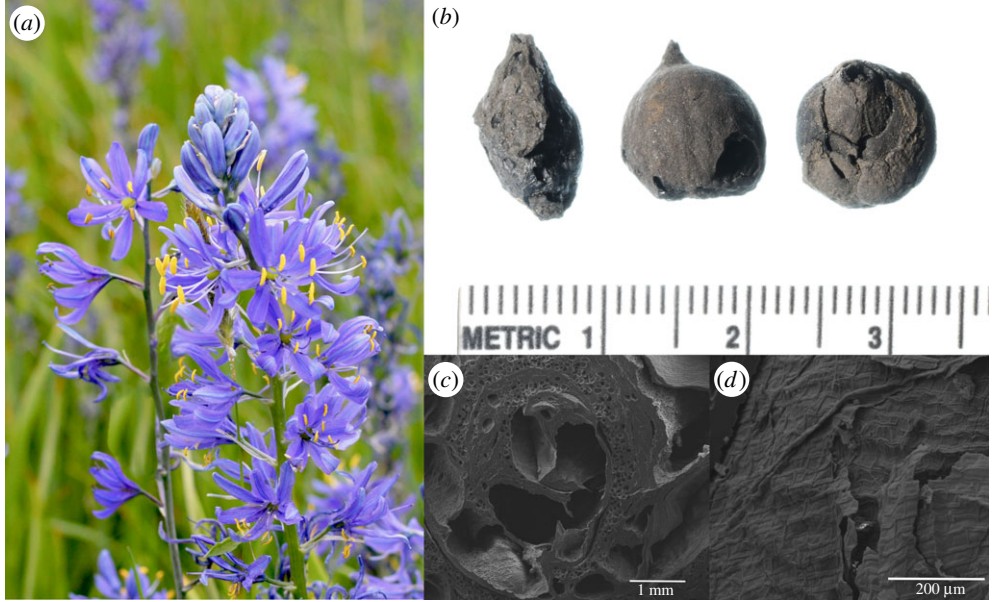

**Figure 2.** Common camas bulbs illustrating main features in the text. (*a*) Camas in bloom. (*b*) Example of whole bulbs dated to 1828–1710 cal BP. Units in centimetres. (*c*) Scanning electron micrograph of a bulb cross-section showing heat-fused leaf scales. (*d*) Scanning electron micrograph of common camas epidermis.

**Table 1.** Camas harvesting strategies, bulb maturities, archaeological evidence and human labour and resource conservation considerations.

| strategy | life cycle stage at harvest | archaeological evidence | implications |
|---|---|---|---|
| 'stripping' | bulbs across all ages/maturity collected | wide range of leaf scales and bulb sizes or larger proportion of immature bulbs (≤3 leaves) | lower associated time costs, may deplete resources locally |
| 'selective harvesting' | predominantly sexually mature camas bulbs, assume immature plants are replanted | primarily mature leaf scale numbers (4 + leaves) | greater time inputs at harvest, but actions ensure future harvests |

Based on the previous literature review summarizing research on camas life cycles, previous camas growth experiments, ethnohistoric sources and contemporary traditional camas knowledge, we predict that past peoples engaged in strategic harvesting practices that involved harvesting only the sexually mature bulbs and replanting immature bulbs for sustainable fields and future harvests (table 1). We further hypothesize that human selection may have resulted in a reduction in time to maturity and/or increase in bulb size. To evaluate these hypotheses, we compared the leaf scales of archaeological camas bulbs from five riverine sites in the Pend Oreille Valley of northeastern Washington State, United States, with those from a comparative collection of modern, experimentally grown mature and immature camas bulbs. Below, we report these data along with 39 existing and two new radiocarbon dates (electronic supplementary material, dataset S1, and figure S1) to evaluate bulb harvests across time and climatic conditions.

# 3. Methods

## 3.1. Archaeological materials

The materials analysed in this study include 110 carbonized *Camassia quamash* bulbs from five sites along the Pend Oreille River in Kalispel ancestral lands, northeastern Washington State (figure 1). Ethnographic, archaeological and historical ecological data from the Pend Oreille Valley, WA indicate

this valley was among the most renowned camas harvesting grounds among interior northwest groups, with groups travelling from afar to collect camas [39,40]. Most of the analysed bulbs were part of legacy collections associated with four sites salvage excavated during the 1985 and 1987 Calispell Valley Archeological Project (CVAP) field seasons [41]. Bulbs from a fifth site (45PO435) were recovered in 2014 and 2015 field seasons excavations conducted by Eastern Washington University and Washington State University [42]. The bulbs selected for analysis were recovered from well-provenanced archaeological features with closely associated radiocarbon dates including earth ovens, hearths, house floors and one special-use structure (electronic supplementary material, dataset S1). All radiocarbon dates were recalibrated using OxCal 4.4.1 with the IntCal 20 curve (electronic supplementary material, figure S1); median calibrated dates are used in the rest of this analysis [43,44].

## 3.2. Palaeoethnobotanical methods

We examined over 100 modern, nursery-grown *Camassia quamash* bulbs of known ages for comparative data on common camas life cycles and ages (electronic supplementary material, dataset S1) [45]. Fifty specimens each of mature bulbs (greater than 4 years of age) and immature bulbs (2–3 years of age) were measured, including bulb length, width and thickness to approximate bulb volume as well as bulb weight. Bulbs were measured fresh shortly after harvesting during summer. They were then wrapped in aluminium foil and placed within the sand to create an anaerobic environment similar to earth ovens and experimentally charred in a muffle furnace at 350°C for 1 h. Bulbs were remeasured, and 30 of each group of modern bulbs were then dissected and the number of leaf scales counted.

Archaeological, carbonized bulbs were identified as *Camassia quamash* by Stenholm [40] and confirmed by the lead author (electronic supplementary material, dataset S2). For all 108 archaeological bulbs, we measured length, width and thickness, as well as estimated bulb preservation, and counted the number of visible leaf scales for each of the carbonized bulbs. Bulb length, width and thickness were recorded to the 0.01 mm using a Neiko stainless steel digital caliper. Weight was recorded to the 0.001 g using a digital scale. Bulb preservation was estimated based upon fragmentation (i.e. whole bulbs without any breakage were deemed 100% preserved). When bulbs included some breakage (i.e. less than 100% preserved), it was possible to examine the basal plate or broken emergent scape to record the number of leaf scales present. To preserve material for future analyses, we randomly subsampled 20% of all bulbs, dissected a cross-section, and counted the number of leaf scales.

We also calculated a simple ratio to approximate bulb size. Camas bulbs vary between 'pencil-like' to teardrop shaped to spherical [28], making bulb volume calculations difficult. It is also estimated that 40% of bulb weight is lost during steaming preparation and an additional 25% lost during charring, though general bulb shape is preserved [46]. Given these taphonomic considerations, we employ a ratio of bulb diameter to bulb length to establish a proxy for bulb size.

$$\frac{(\text{width} + \text{thickness})/2}{\text{length}}.$$

All archaeological bulbs with leaf scale data and associated with radiocarbon-dated features were binned into 500-year increments. Leaf scale data were graphically plotted to assess similarity and differences in archaeological and modern camas samples. These relationships were tested in R v. 4.0.2 using the Mann–Whitney U test with the Benjamini and Hochberg [47–49] *p*-value adjustment for multiple comparisons and the Kruskal–Wallis rank-sum test [50], a non-parametric equivalent to ANOVA. All statistical significance inferences are made via comparison with a 0.05 confidence level and calculated effect sizes.

# 4. Results

## 4.1. Proxies for maturity

For the modern comparative camas bulbs, our results confirm previous findings [28,30] that bulb dimensions varied widely and were not adequate proxies for bulb maturity (charred size ratio: $U = 113.5$, $p = 1.128 \times 10^{-6}$). The quantity of interior bulb leaf scales, however, were significantly different between mature and immature samples ($U = 60$, $p = 1.186 \times 10^{-9}$). Based upon these comparative data we conclude that bulb leaf scales are an excellent approximation of relative bulb age, as sexually immature bulbs have fewer leaves (3 or less) and sexually mature bulbs have more leaves (4–5 or more).

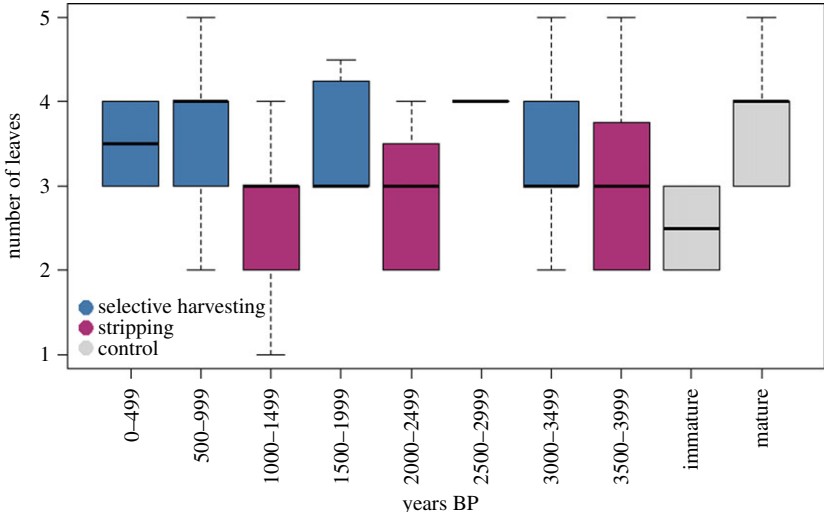

**Figure 3.** Comparisons of camas bulb leaf scale data by period.

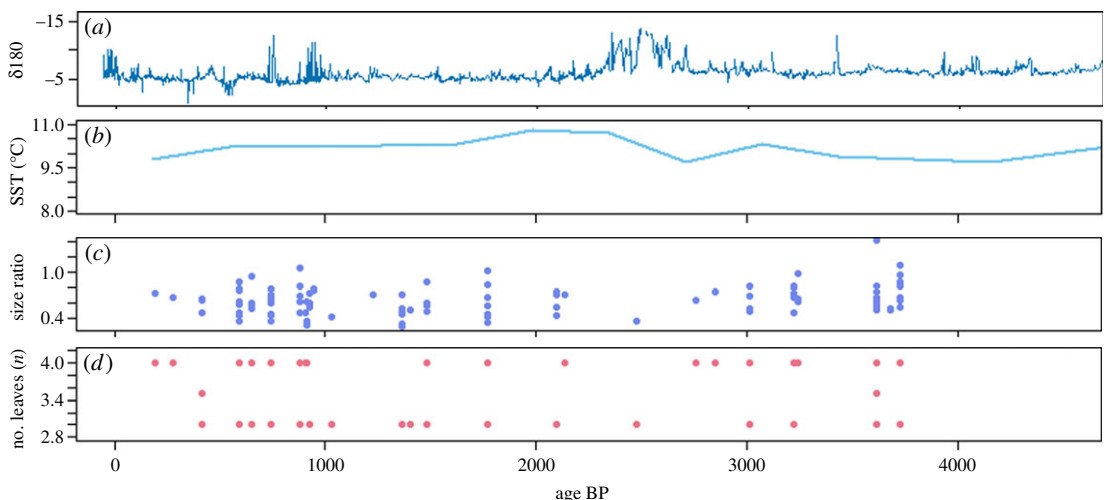

**Figure 4.** Summary of Pend Oreille Valley human–camas relationships through time (dates reported in cal BP). (*a*) $\delta$18O data from Cleland lake, British Columbia [51]. (*b*) Sea surface temperature reconstruction from Vancouver Island (°C) [52]. (*c*) Median size ratio for bulbs by median date (cal BP). (*d*) Number of leaves (*n*) by median date (cal BP).

## 4.2. Camas harvesting strategies

We found that people alternated between two camas harvesting strategies across the last 4000 years (figure 3). Archaeological leaf scale data for the oldest period, 3999–3500 cal BP, have a large range. The Mann–Whitney $U$ test indicates bulbs are different from control mature bulb populations ($U = 148$, $p = 0.01$, $\eta^2 = 0.181$) but with size ratios that are comparable to the size ratios of other periods (figure 4*c*, see below). We see this period as a time in which people opportunistically collected camas bulbs regardless of maturity (i.e. 'stripping').

A second harvesting strategy, 'selective harvesting,' is evident across several periods including the 3499–3000, 1999–1500 and 999–present periods. Comparisons between the archaeological leaf sample distributions are different from the immature control data, with strong effect sizes ($\eta^2 > 0.2$) and are statistically indistinguishable at the 0.05 confidence level from the mature control population (electronic supplementary material, table S1). These data indicate that most camas bulbs harvested during these periods were sexually mature bulbs. We infer that most, if not all, immature bulbs were replanted or not harvested.

The data for 2499–1000 cal BP indicate oscillations between harvesting practices. For both the 2499–2000 and 1499–1000 cal BP periods, archaeological leaf scale data indicate that people were much more likely to harvest bulbs across all maturity levels (mean leaf scales of 2.875 and 2.654, respectively).

**Table 2.** Presence and absence of vegetative crop domestication syndrome traits within *Camassia* spp. after Denham *et al.* [3]. Denham *et al.* [3, p. 589] consider bulbous underground storage organs as primarily propagated vegetatively and here we follow their criteria for domestication in asexually propagated plants.

| trait category | domestication in asexually propagated plants | modern camas traits | archaeological camas trait presence |
|---|---|---|---|
| mode of reproduction | 1. partial or complete loss of sexual reproduction ability<br>2. increased uniformity in clonal reproduction traits | 1. estimated 15% reproduce asexually [37]<br>2. unknown | 1. unknown<br>2. unknown |
| plant life cycle | shift towards biennial to annual life cycle | perennial life cycle | perennial life cycle; no evidence for decrease in time to maturity (this study) |
| yield of edible portion | increased size | no evidence [28]; potential to increase size over several generations [29] | no evidence for change in size over time [30], (this study) |
| ease of harvesting | development of easily separated underground storage organs | easily harvested with digging sticks; many bulbs are easily separated [53] | unknown, but presence in many archaeological sites indicates easily harvested |
| timing of production | asynchronous and more continuous production, with in-ground storage | usually harvested in summer, but can be harvested in winter as well [53] | unknown; ethnohistoric evidence of multi-season harvests; potential to look at bulb phenology [20] |
| environmental tolerance | traits that enable cultivation across wider environmental range | popular species among gardeners and bulb enthusiasts; easily transplanted in restoration contexts | no evidence for human dispersal [31]; archaeological presence beyond modern range [25] interpreted as trade; historically recorded beyond contemporary range [4] |

Archaeological sample distributions are significantly different from the control mature camas bulb populations at the 0.05 confidence level (electronic supplementary material, table S1) and large effect size values ($\eta^2 > 0.18$). To contrast this, the 1999–1500 cal BP period contains bulbs from a single feature that average 3.5 leaf scales. For this period, comparisons between the archaeological bulb sample distributions are different from the immature control data ($p = 0.011$, $\eta^2 = 0.23$) and indicate harvesting of older bulbs.

## 4.3. Evidence for management

To assess for selection within carbonized bulb storage organs, we also tested ratio and weight variables to assess change in bulb size relative to bulb maturity. The Kruskal–Wallis test indicates that bulb size does not vary significantly throughout time ($\chi^2 = 9.806$, $p = 0.279$, $\varepsilon^2 = 0.088$), nor does bulb weight ($\chi^2 = 7.573$, $p = 0.376$, $\varepsilon^2 = 0.223$). We conclude that there is no evidence to support our second hypothesis that Pend Oreille Valley camas bulbs were selected for an increase in size or weight, nor for a reduction in time to maturity. Furthermore, as bulb size and weight do not vary significantly through time, we see no supporting evidence for climatic influences on camas bulb size.

We further compared modern and archaeological camas traits to domestication syndrome traits in asexually propagated geophytes (table 2) [3]. There are seven possible vegetative crop domestication syndrome traits for *Camassia* spp. We see supporting evidence for three traits: camas is easily

harvested and can be harvested across multiple seasons, with about 15% reproducing asexually through offshoot bulblets. We are not aware of any further research on camas clonal reproduction, nor did we find evidence for an increase in size over time, size standardization or decrease in time to maturity. While one study found no evidence for human-mediated translocation across a small portion of the Northwest region [31], ethnohistoric records and contemporary restoration efforts indicate camas bulbs are easily transplanted and propagated [28,54]. Given this variation in phenotypic traits in archaeological and modern camas, we infer that the traits in table 2 are the result of long-term management and human–camas entanglement rather than biological selection.

# 5. Discussion and conclusion

## 5.1. Climate and camas consumption

Plateau plant food syntheses often look to climatic patterns in contextualizing cultural change [37,55], and in this section, we follow this tradition by comparing our results with palaeoclimatological datasets. By at least 4000 years ago, geophytes in general become an integral part of the diet for residents of the Columbia Plateau [55], and archaeological evidence indicates camas was first processed on a large scale within the Pend Oreille Valley at this time [37]. The earliest dated material in our study corresponds with a climatic period of pronounced seasonality (i.e. cooler, wetter winters than present day) as determined by lake modelling analyses and $\delta^{18}$O records [51]. As camas plants thrive with cooler winters and warm and dry summers, it is possible that this period is correlated with an abundance of camas and expansion of productive habitat. We see two possible interpretations for the 3999–3500 cal BP 'stripped' camas data: either that people had not established social institutions delineating how these plants were harvested, or that they were not strictly following these cultural rules during a period of possible abundance. Plateau archaeological summaries widely accept that practices of bulk-processing camas became fully routine after 3500 cal BP [37,55,56].

In the subsequent period, it appears that residents and visitors to the Pend Oreille Valley adopted a new camas harvesting strategy. Primarily mature bulbs were collected and cooked from 3499–2999 cal BP (i.e. 'selective harvesting'). Thus by approximately 3500 cal BP, it appears people were consciously and deliberately harvesting the bulbs of sexually mature plants, and we infer replanting immature bulbs and seed capsules. These selective harvesting practices exhibit significant care for camas plants and fields, such that these established human–camas relationships productively benefited both plant and human species. Bulbs in this period were also the largest (figure 4c). At roughly the same time, approximately 3000 BP, sea surface temperature data indicate the region experienced and sharp cooling event [52], with $\delta^{18}$O dataset summaries indicating wetter conditions [51]. As camas plants readily tolerate cold temperatures and thrive with winter precipitation, we see no evidence that this globally recognized cold period should negatively affect camas niches or production. Indeed, bulbs in this period were also the largest of our sample (figure 4c,d).

By 2500 cal BP, however, interactions across people, camas and climate changed. At approximately 2200 cal BP, the northern Columbia Plateau climate begins to gradually transition from wetter to drier conditions [51] and our results indicate that people shifted from a focus on harvesting mature plants to vacillating between strategies. In the bulb data from the 2499–2000 and 1499–1000 cal BP periods, leaf scale data are significantly different from that of mature bulbs. We suggest that this evidence of camas bulb 'stripping' may either be due to a relaxation of the social rules governing camas harvests, or the influence of other, unidentified factors influencing human–plant relationships and access to camas fields. The 1999–1500 cal BP data, however, indicate people primarily collected mature bulbs. These data come from a single earth oven feature, and we argue that more data from archaeological camas bulbs are necessary to understand these oscillating harvesting strategies.

Incidentally, this 2499–1000 cal BP period corresponds with widespread cultural changes and a period of stabilized precipitation [51] throughout the Columbia–Fraser Plateau culture area. Intermittently throughout the Columbia–Fraser Plateau, people begin to abandon semi-sedentary lifestyles for more mobile patterns of movement. In many places, pithouses, the architectural residence of choice for 3000 years prior, are abandoned in favour of lightweight mat lodges [57–59]. Rock shelters were repurposed from habitation sites to storage caches [60]. Trade and human movement increased throughout the region [61], and on the coast, there is evidence for the trade of camas across hundreds of kilometres [25]. These significant material culture changes most likely reflect other dramatic changes in everyday life.

By 1000 cal BP, people return to primarily harvesting mature bulbs. This 'selective harvesting' pattern continues through to the contact period and is probably the pattern from which our ethnographic sources for plant management and stewardship arise. Despite global and regional evidence for the Little Ice Age cooling event, archaeological activity does not diminish during this period for the valley [41], nor are there significant changes in bulb maturity, size or weight at harvest. We suggest that the cultural institutions surrounding camas harvests ensured ongoing, sustainable harvests despite a subtle decrease in regional precipitation throughout the late Holocene [51,62] and global cooling events such as the Little Ice Age [63]. Indeed, throughout our 4000-year study period, we see little correspondence between palaeoclimates and harvesting strategies as well as no evidence for change in bulb size or weight due to environmental conditions or other forms of biological selection.

## 5.2. Management without selection

While many who work closely with camas suggest that it may have been domesticated in some form, our research supports other studies that do not find unequivocal evidence for selection or human-aided dispersal patterns [30,31]. Contemporary domestication explanations see the repetitive cycle of sowing, collecting and resowing of wild plants as the primary processes through which novel phenotypic and genotypic traits emerge [64], yet here we have a case in which such human behaviours and intimate knowledge of plant life cycles do not act as selective forces. These long-term entanglements or interactions should not be seen as incipient agriculture, but instead flexible relationships with plants that resulted in versatile yet sustainable subsistence practices.

There is increasing evidence that across the Americas, societies consistently maintained mixed subsistence strategies for millennia, relying partially on the cultivation of domesticated plants or management of non-domesticates, and that many of these practices do not result in large-scale food production [8,22,65]. Leading scholars [5,8,15] now argue that agriculture is not an inevitable product of domestication. We further add that domestication is not an automatic outcome of management or extensive niche construction activities [6]. As scholarly dialogue continues to examine the roles and influences plants have upon people and the entangled relationships that ensue [9,11], studies that prioritize human–plant relational interactions offer fresh frameworks for theorizing and examining alternative modes of production and the development of stable yet dynamic non-agricultural subsistence systems.

## 5.3. Traditional ecological knowledge informs the past and the future

This study establishes patterns of camas harvesting strategies and behaviours for one valley within the North American Columbia Plateau from 4000 years ago to the post-contact period. The identified 'selective harvesting' practices over the past 1000 years support ethnographic descriptions of camas harvests throughout the Northwest and supplant our knowledge of how such behaviours may have changed over time. This result establishes considerable continuity to human intervention into camas life cycles. Furthermore, these results support Indigenous oral traditions and contemporary practices of selective harvesting and plant food resource stewardship, extending traditional ecological knowledge systems at least 3500 years back into time. In future studies, we plan on extending this methodology to archived archaeological collections of camas bulbs throughout the Northwest, as well as looking into the phenology or seasonal timing of past camas harvests. We may also use these selective harvesting insights into human–camas relationships to model contemporary camas management and harvests, ensuring sustainable food options for Indigenous communities working to restore and reclaim autonomy over their health, well-being and cultural heritage [66–69].

We also demonstrate that for camas and other perennial geophyte plants, maturity at harvest is not synonymous with plant organ size, and we argue maturity has more utility in researching questions of cultivation, domestication or management. We suggest that others investigating human–geophyte interactions and geophyte domestication pathways consider plant life-history traits and time to sexual or asexual maturity in future studies. This approach offers an alternative explanatory framework to conventional management studies and can be applied to other vegetatively propagated species. Such a perspective provides significant potential across research agendas spanning forager–plant relationships through time, the advent of cultivation practices, and in teasing out varied management/domestication pathways.

Ethics. We submitted and were approved a research request form with the Tribe's Cultural Resources Department and accessed the curated archaeological material at the Kalispel Tribe's curational facility with full permission.

Data accessibility. All data used in the above analyses are archived at https://doi.org/10.5281/zenodo.4562953. All notes, artefacts and samples from the project are curated at the Kalispel Tribe's curational facility in Usk, Washington State.

Authors' contributions. M.C. and J.d.A.G. conceived of and designed the study. M.C. and T.M. completed the bulb measurements. All authors (M.C., S.T., T.M. and J.d.A.G.) helped with the data analysis as well as drafting and revising the manuscript. All authors also approved the final publication and agree to be held accountable for this work.

Competing interests. The authors declare no competing interests.

Funding. This work was supported by the Association for Washington Archeology Student Travel Grant, the Washington State University Graduate and Professional Student Association Dissertation Grant, the Washington State University College of Arts and Sciences Boeing Fellowship in Environmental Studies and the University of California, San Diego.

Acknowledgements. We are especially grateful to the Kalispel Tribe for allowing us to work with and understand the history of their ancestors. We thank Natasha Lyons, Henry Hooghiemstra, Bill Andrefsky, Colin Grier and two anonymous reviewers for their insightful comments on this work. Finally, we also wish to thank Stephenie Kramer for inspiring this study, and Cassady Fairlane and Clare Carney for help with figure 1.

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
