## [Peer Review File · Royal Society Open Science]

Review History

RSOS-202213.R0 (Original submission)

Review form: Reviewer 1 (Natasha Lyons)

Is the manuscript scientifically sound in its present form?

Yes

Are the interpretations and conclusions justified by the results?

Yes

Is the language acceptable?

Yes

Do you have any ethical concerns with this paper?

No

Have you any concerns about statistical analyses in this paper?

Yes

Recommendation?

Accept with minor revision (please list in comments)

Comments to the Author(s)

This is an excellent manuscript, Molly et al. Well done! I have minor comments throughout and some suggested tweaks and references (see Appendix A).

Review form: Reviewer 2 (Mana Hayashi Tang)**Is the manuscript scientifically sound in its present form?**

Yes

Are the interpretations and conclusions justified by the results?

Yes

Is the language acceptable?

Yes

Do you have any ethical concerns with this paper?

No

Have you any concerns about statistical analyses in this paper?

No

Recommendation?

Accept with minor revision (please list in comments)

Comments to the Author(s)

This is an exciting study for Pacific Northwest paleoethnobotany and broader research into geophytes in the archaeological record. I am supportive of the overall interpretations and conclusions, with a question and minor suggestion to strengthen how you frame them within broader conversations about plant domestication:

In the section where you test your findings against Denham et al.'s (2020) suggested list of domestication syndrome traits in vegetative crops, I am puzzled by the interpretation that lacking evidence on this list suggests that "conscious or unconscious selection" is not involved (p.7, lines 58-60). Do you instead mean that you lack evidence for its role? And do you mean biological selection, as in factors/behaviors that led to "phenotypic and genotypic changes that are different from unmanaged populations" (p.3, lines 6-8)?

Though not necessarily evidence for biological selection, your study adds an important dimension to Denham et al.'s conclusions that phenotypic traits associated with early vegetative crop domestication is not ubiquitous. Maturity at harvest may not be a phenotypic trait associated with evolutionary adaptation, but it is (as you've pointed out) a physically recognizable variable for measuring the developmental stage of a geophyte in paleoethnobotanical evidence. By proxy, it can point to selective management practices. In a sense, your study is a strong case for how whether camas were domesticated or not is beside the point - it does not diminish their entanglement with Indigenous communities in the Northwest. In light of your conclusions on p.9, I would suggest to lean more heavily into the significance of

seeing "considerable continuity to human intervention into camas life cycles" (p.1, lines 38-39), when assessing your findings against Denham et al.'s list, and domestication studies in general.

Again, this is incredible research - congratulations! I look forward to seeing it published.

Review form: Reviewer 3

Is the manuscript scientifically sound in its present form?

Yes

Are the interpretations and conclusions justified by the results?

Yes

Is the language acceptable?

Yes

Do you have any ethical concerns with this paper?

No

Have you any concerns about statistical analyses in this paper?

No

Recommendation?

Accept with minor revision (please list in comments)

Comments to the Author(s)

General comments to authors:

In this manuscript, the authors present an interesting, well-planned, and generally well-executed study that shows evidence of changing strategies of camas management on the Columbia Plateau for the past 4000 years. The manuscript is for the most part well written, and I am generally convinced of the argument presented with regard to the camas/archy data and how collecting/management strategies seem to have changed through time.

With that said, I find the linkages presented between the camas data/archy data and the paleoclimate data a lot less compelling. I am unconvinced that the paleoclimatic data presented in the study actually apply to this study area. I am not saying they don't, but this isn't clear from the manuscript as it reads now. In particular, there needs to be a better explanation of what the SST and d18O records are saying about past climate on the Columbia Plateau. Much of this might be accomplished by providing a more detailed caption for Figure 4. Once this is made clear, then the paleoclimatic data and the camas data need to be integrated in a more meaningful way so that the climatic changes discussed bear some relevance for the camas data. Right now it seems like the paleoclimatic information and the camas data are standing beside each other, but are not really talking to each other or showing any real linkages (see more specific comments below). Lastly, I believe that the camas data would be more honestly represented in Figure 4 by bars than a time series because you binned the data into 500-year intervals. By having the data as single points connected by straight lines, it implies that there are increasing and decreasing trends between those data points throughout the time they span, and in reality this may or may not be true (but your data don't have the resolution needed to say one way or another). You should represent the camas data in figure 4 as bars (like a histogram) that span the correct time periods.

My only other substantive comment is that you are very inconsistent with how you refer to time throughout the manuscript, in particular in the Results and Discussion sections. Cal yr BP, BP, cal BP, years, years ago. Go through and make sure these all match, especially when referring to calibrated radiocarbon years (“cal yr BP” is standard).

Specific comments to authors:

1. Summary

Page 2 Line 31: You say in the summary that you “bring together archaeological, paleoecological, and botanical data”, but yet I don’t really see any paleoecological data (for instance, pollen data) mentioned in the paper. I think perhaps you mean “paleoclimatological” data? Probably best to change the wording. I was expecting from the intro that you were going to discuss vegetation reconstructions for the late Holocene, but you didn’t. This might be a good idea, however... (but likely beyond the scope of this paper).

Line 34: Need to make it clear at the start of the sentence, “Throughout the mid to late Holocene...” that you are talking about your results from this study. Further, I would characterize the past 4000 years as just the “late Holocene.”

Line 38: Can you be more specific about what you mean by “other domestication syndrome traits”? Maybe list them here in the summary.

2. Introduction

Lines 53 and 54: Need a space between end of sentence and citation.

Page 3 Lines 1-4: This is kind of a long and convoluted sentence. Can you shorten or break into multiple sentences to improve clarity?

Line 4: “these alternative pathways” is a bit vague. Please clarify what you are talking about here.

Line 17: Wouldn’t you say it is both “conscious and unconscious decisions and behaviors”?

Think about including that word, too.

Line 28: Space needed before citation.

Line 29: Change “which” to “that.”

3. Human Relationships with Camas Life Cycles

Page 4 Line 26: I think it would be better to say “seasonally xeric moisture regimes.” Consider revising.

Page 5 Line 10: Citation 36 is incomplete in the bibliography.

Line 25: Comma needed after “knowledge.”

Line 26: Probably best to remove the comma and insert the word “and” before “replanting.”

4. Methods

Page 6 Line 25: I guess I am left wondering why you didn’t just count them all. From what I can tell, it doesn’t seem like that much work so there should be some sort of explanation why you didn’t analyze all of the samples.

Lines 33-35: This looks strange to me. I would just write out the equation verbally in a sentence.

5. Results

Lines 54-55: It took me a while to figure out whether your first sentence here is summarizing the literature or stating what you learned from your study. Please make it more clear that you are stating this based on your results. Also somewhat confusing is the clause “throughout the study period.” Consider revising.

Line 56: The statement “bulbs are different than both the mature bulb populations” is confusing. Are you referring to the control population? Please clarify.

Line 60: I suggest inserting “i.e.,” before “stripping” in the parentheses. Same in other instances throughout manuscript.

Page 7 Line 0: Need to make it clear here that you are talking about strategy (i.e., selective harvesting) in this sentence. Add language to indicate.

Lines 0-1: Rewrite to say “ 3499-3000 BP, 1999-500 BP, and 999 BP-present periods.” Be consistent.

Line 5: Remove semi-colon and start new sentence at “We infer the most...”

Line 32: Remove the word “temporal.”

Line 34: Comma needed before “respectively.”

Line 35: Are you referring to the control population? Please clarify.

Line 36: End parenthesis missing. Need "BP" after "1999-500." Delete "temporal."

Line 37: Add comma after "For the period,".

Line 47-48: You say you "see no evidence for climatic influences on camas bulb size." On what are you basing this conclusion? If you are going to state this here, then it needs some context.

Table 2: Check weird superscript "7" in fourth column.

6. Discussion

Page 8 Line 44-45: How do you know this is a period of "pronounced seasonality"? Need to include a brief description of where the data come from.

Lines 48-50: This is a poorly written sentence and it makes it so the point you are trying to get across is not as strong as it could be. Rewrite to remove passive voice in both clauses.

Lines 50-51: You need to make it clear how you know this. Is this based on the archaeological evidence, or is this an interpretation based on your study results? Language needs to be softened to make it seem less of a certainty and more of a likelihood that this happened based on what your data show.

Line 53: "temporal period" is redundant. Please delete "temporal" here and in other places in the text.

Line 53: Explain how you know that it was "residents and visitors" or revise language.

Lines 55-57: Again, this statement of "only" is a bit strong. Revise to indicate that this is what seems likely based on what your data show, but that you cannot know for certain.

Lines 60- page 9 Line 1: You need to find a way to better weave together the paleoclimatic and camas data in this section. Instead of having separate sentences about climate and camas, you need to blend the ideas together to make a stronger statement. Also, if the "globally recognized cold period" (which honestly I don't really think you have the data to show this since your SST record is from the NE Pacific), why mention it at all? Please rewrite this section.

Line 9: Be consistent with how you report years.

Line 10: Get rid of passive voice, "and there follows a 500-year period which primarily mature bulbs were collected."

Line 15: Cultural "changes"? If so, indicate.

Lines 15-26: Watch the tenses used in this paragraph. You switch back and forth several times.

Lines 22-26: Again, the climatic description seems somewhat awkwardly thrown into this paragraph without providing any real context for how it applies to your data. What would these oscillations mean for camas or for people's efforts to harvest it? If it is relevant, then you need to explain how. You also need to explain where you are getting the data for these oscillations ("as known from..."). The last sentence of this paragraph seems like a waste, and I would argue that it isn't further climatic resolution that is needed, but a way to better integrate these data sets given their vastly different sampling resolutions. Rewrite this section to better indicate how the observed climatic shifts either would or would not have influenced camas growth and management.

Line 28: Again, make it clear that you are basing what you are saying on your data.

Line 31: What do you mean by "archaeological evidence does not subside"? Please clarify.

Line 34: On what are you basing your statement of "a subtle decrease in regional precipitation throughout the late Holocene"? Are you referring to SST data? If so, I don't really see that in the data shown in Figure 4. The resolution of the record shown in panel B is much too coarse of a resolution to indicate that. If you are saying it's shown by the $\delta^{18}O$ record, then you need to make that clear.

Page 10 Line 7: It seems strange to bring up "economic systems" here, considering that's not really the focus of your study. Consider revising.

Line 16: Change "which" to "that."

Line 23: Several of your results? Be clear and provide examples.

Line 27: Revise so as not to repeat the use of the word "also" here, earlier in this paragraph, and at the start of the next paragraph.

Lines 37-40: This statement seems like a stretch to me. I suggest deleting it or scaling it back.

Review form: Reviewer 4

Is the manuscript scientifically sound in its present form?

Yes

Are the interpretations and conclusions justified by the results?

Yes

Is the language acceptable?

Yes

Do you have any ethical concerns with this paper?

No

Have you any concerns about statistical analyses in this paper?

No

Recommendation?

Accept with minor revision (please list in comments)

Comments to the Author(s)

Please see the attached file (Appendix B).

Decision letter (RSOS-202213.R0)

Dear Dr Carney

On behalf of the Editors, we are pleased to inform you that your Manuscript RSOS-202213 "Harvesting strategies as evidence for 4,000 years of camas (*Camassia quamash*) management in the North American Columbia Plateau" has been accepted for publication in Royal Society Open Science subject to minor revision in accordance with the referees' reports. Please find the referees' comments along with any feedback from the Editors below my signature.

Please submit your revised manuscript and required files (see below) no later than 7 days from today's (ie 19-Feb-2021) date. Note: the ScholarOne system will 'lock' if submission of the revision is attempted 7 or more days after the deadline. If you do not think you will be able to meet this deadline please contact the editorial office immediately.

Please note article processing charges apply to papers accepted for publication in Royal Society Open Science (<https://royalsocietypublishing.org/rsos/charges>). Charges will also apply to

papers transferred to the journal from other Royal Society Publishing journals, as well as papers submitted as part of our collaboration with the Royal Society of Chemistry (<https://royalsocietypublishing.org/rsos/chemistry>). Fee waivers are available but must be requested when you submit your revision (<https://royalsocietypublishing.org/rsos/waivers>).

Best regards,

on behalf of Professor Leslie Brown (Associate Editor) and Kevin Padian (Subject Editor)
openscience@royalsociety.org

Associate Editor Comments to Author (Professor Leslie Brown):

Please ensure that you have addressed all the comments/suggestions made by the reviewers to ensure a sound scientific paper.

Reviewer comments to Author:

Reviewer: 1
Comments to the Author(s)

This is an excellent manuscript, Molly et al. Well done! I have minor comments throughout and some suggested tweaks and references (see attached file 'RSOS-202213_Proof +NL.pdf').

Reviewer: 2
Comments to the Author(s)

This is an exciting study for Pacific Northwest paleoethnobotany and broader research into geophytes in the archaeological record. I am supportive of the overall interpretations and conclusions, with a question and minor suggestion to strengthen how you frame them within broader conversations about plant domestication:

In the section where you test your findings against Denham et al.'s (2020) suggested list of domestication syndrome traits in vegetative crops, I am puzzled by the interpretation that lacking evidence on this list suggests that "conscious or unconscious selection" is not involved (p.7, lines 58-60). Do you instead mean that you lack evidence for its role? And do you mean biological selection, as in factors/behaviors that led to "phenotypic and genotypic changes that are different from unmanaged populations" (p.3, lines 6-8)?

Though not necessarily evidence for biological selection, your study adds an important dimension to Denham et al.'s conclusions that phenotypic traits associated with early vegetative crop domestication is not ubiquitous. Maturity at harvest may not be a phenotypic trait associated with evolutionary adaptation, but it is (as you've pointed out) a physically recognizable variable for measuring the developmental stage of a geophyte in paleoethnobotanical evidence. By proxy, it can point to selective management practices. In a sense, your study is a strong case for how whether camas were domesticated or not is beside the

point - it does not diminish their entanglement with Indigenous communities in the Northwest. In light of your conclusions on p.9, I would suggest to lean more heavily into the significance of seeing "considerable continuity to human intervention into camas life cycles" (p.1, lines 38-39), when assessing your findings against Denham et al.'s list, and domestication studies in general.

Again, this is incredible research - congratulations! I look forward to seeing it published.

Reviewer: 3

Comments to the Author(s)

General comments to authors:

In this manuscript, the authors present an interesting, well-planned, and generally well-executed study that shows evidence of changing strategies of camas management on the Columbia Plateau for the past 4000 years. The manuscript is for the most part well written, and I am generally convinced of the argument presented with regard to the camas/archy data and how collecting/management strategies seem to have changed through time.

With that said, I find the linkages presented between the camas data/archy data and the paleoclimate data a lot less compelling. I am unconvinced that the paleoclimatic data presented in the study actually apply to this study area. I am not saying they don't, but this isn't clear from the manuscript as it reads now. In particular, there needs to be a better explanation of what the SST and $\delta^{18}O$ records are saying about past climate on the Columbia Plateau. Much of this might be accomplished by providing a more detailed caption for Figure 4. Once this is made clear, then the paleoclimatic data and the camas data need to be integrated in a more meaningful way so that the climatic changes discussed bear some relevance for the camas data. Right now it seems like the paleoclimatic information and the camas data are standing beside each other, but are not really talking to each other or showing any real linkages (see more specific comments below). Lastly, I believe that the camas data would be more honestly represented in Figure 4 by bars than a time series because you binned the data into 500-year intervals. By having the data as single points connected by straight lines, it implies that there are increasing and decreasing trends between those data points throughout the time they span, and in reality this may or may not be true (but your data don't have the resolution needed to say one way or another). You should represent the camas data in figure 4 as bars (like a histogram) that span the correct time periods.

My only other substantive comment is that you are very inconsistent with how you refer to time throughout the manuscript, in particular in the Results and Discussion sections. Cal yr BP, BP, cal BP, years, years ago. Go through and make sure these all match, especially when referring to calibrated radiocarbon years ("cal yr BP" is standard).

Specific comments to authors:

1. Summary

Page 2 Line 31: You say in the summary that you "bring together archaeological, paleoecological, and botanical data", but yet I don't really see any paleoecological data (for instance, pollen data) mentioned in the paper. I think perhaps you mean "paleoclimatological" data? Probably best to change the wording. I was expecting from the intro that you were going to discuss vegetation reconstructions for the late Holocene, but you didn't. This might be a good idea, however... (but likely beyond the scope of this paper).

Line 34: Need to make it clear at the start of the sentence, "Throughout the mid to late Holocene..." that you are talking about your results from this study. Further, I would characterize the past 4000 years as just the "late Holocene."

Line 38: Can you be more specific about what you mean by "other domestication syndrome traits"? Maybe list them here in the summary.

2. Introduction

Lines 53 and 54: Need a space between end of sentence and citation.

Page 3 Lines 1-4: This is kind of a long and convoluted sentence. Can you shorten or break into multiple sentences to improve clarity?

Line 4: "these alternative pathways" is a bit vague. Please clarify what you are talking about here.

Line 17: Wouldn't you say it is both "conscious and unconscious decisions and behaviors"?

Think about including that word, too.

Line 28: Space needed before citation.

Line 29: Change "which" to "that."

3. Human Relationships with Camas Life Cycles

Page 4 Line 26: I think it would be better to say "seasonally xeric moisture regimes." Consider revising.

Page 5 Line 10: Citation 36 is incomplete in the bibliography.

Line 25: Comma needed after "knowledge."

Line 26: Probably best to remove the comma and insert the word "and" before "replanting."

4. Methods

Page 6 Line 25: I guess I am left wondering why you didn't just count them all. From what I can tell, it doesn't seem like that much work so there should be some sort of explanation why you didn't analyze all of the samples.

Lines 33-35: This looks strange to me. I would just write out the equation verbally in a sentence.

5. Results

Lines 54-55: It took me a while to figure out whether your first sentence here is summarizing the literature or stating what you learned from your study. Please make it more clear that you are stating this based on your results. Also somewhat confusing is the clause "throughout the study period." Consider revising.

Line 56: The statement "bulbs are different than both the mature bulb populations" is confusing. Are you referring to the control population? Please clarify.

Line 60: I suggest inserting "i.e.," before "stripping" in the parentheses. Same in other instances throughout manuscript.

Page 7 Line 0: Need to make it clear here that you are talking about strategy (i.e., selective harvesting) in this sentence. Add language to indicate.

Lines 0-1: Rewrite to say "3499-3000 BP, 1999-500 BP, and 999 BP-present periods." Be consistent.

Line 5: Remove semi-colon and start new sentence at "We infer the most..."

Line 32: Remove the word "temporal."

Line 34: Comma needed before "respectively."

Line 35: Are you referring to the control population? Please clarify.

Line 36: End parenthesis missing. Need "BP" after "1999-500." Delete "temporal."

Line 37: Add comma after "For the period,".

Line 47-48: You say you "see no evidence for climatic influences on camas bulb size." On what are you basing this conclusion? If you are going to state this here, then it needs some context.

Table 2: Check weird superscript "7" in fourth column.

6. Discussion

Page 8 Line 44-45: How do you know this is a period of "pronounced seasonality"? Need to include a brief description of where the data come from.

Lines 48-50: This is a poorly written sentence and it makes it so the point you are trying to get across is not as strong as it could be. Rewrite to remove passive voice in both clauses.

Lines 50-51: You need to make it clear how you know this. Is this based on the archaeological evidence, or is this an interpretation based on your study results? Language needs to be softened to make it seem less of a certainty and more of a likelihood that this happened based on what your data show.

Line 53: "temporal period" is redundant. Please delete "temporal" here and in other places in the text.

Line 53: Explain how you know that it was "residents and visitors" or revise language.

Lines 55-57: Again, this statement of “only” is a bit strong. Revise to indicate that this is what seems likely based on what your data show, but that you cannot know for certain.

Lines 60- page 9 Line 1: You need to find a way to better weave together the paleoclimatic and camas data in this section. Instead of having separate sentences about climate and camas, you need to blend the ideas together to make a stronger statement. Also, if the “globally recognized cold period” (which honestly I don’t really think you have the data to show this since you SST record in from the NE Pacific), why mention it at all? Please rewrite this section.

Line 9: Be consistent with how you report years.

Line 10: Get rid of passive voice, “and there follows a 500-year period which primarily mature bulbs were collected.”

Line 15: Cultural “changes”? If so, indicate.

Lines 15-26: Watch the tenses used in this paragraph. You switch back and forth several times.

Lines 22-26: Again, the climatic description seems somewhat awkwardly thrown into this paragraph without providing any real context for how it applies to your data. What would these oscillations mean for camas or for people’s efforts to harvest it? If it is relevant, then you need to explain how. You also need to explain where you are getting the data for these oscillations (“as known from...”). The last sentence of this paragraph seems like a waste, and I would argue that it isn’t further climatic resolution that is needed, but a way to better integrate these data sets given their vastly different sampling resolutions. Rewrite this section to better indicate how the observed climatic shifts either would or would not have influenced camas growth and management.

Line 28: Again, make it clear that you are basing what you are saying on your data.

Line 31: What do you mean by “archaeological evidence does not subside”? Please clarify.

Line 34: On what are you basing your statement of “a subtle decrease in regional precipitation throughout the late Holocene”? Are you referring to SST data? If so, I don’t really see that in the data shown in Figure 4. The resolution of the record shown in panel B is much too coarse of a resolution to indicate that. If you are saying it’s shown by the $\delta^{18}O$ record, then you need to make that clear.

Page 10 Line 7: It seems strange to bring up “economic systems” here, considering that’s not really the focus of your study. Consider revising.

Line 16: Change “which” to “that.”

Line 23: Several of your results? Be clear and provide examples.

Line 27: Revise so as not to repeat the use of the word “also” here, earlier in this paragraph, and at the start of the next paragraph.

Lines 37-40: This statement seems like a stretch to me. I suggest deleting it or scaling it back.

Reviewer: 4

Comments to the Author(s)

Please see the attached file ('RSOS-202213 rev report').

===PREPARING YOUR MANUSCRIPT===

===PREPARING YOUR REVISION IN SCHOLARONE===

- Ensure that your data access statement meets the requirements at <https://royalsociety.org/journals/authors/author-guidelines/#data>. You should ensure that you cite the dataset in your reference list. If you have deposited data etc in the Dryad repository, please only include the 'For publication' link at this stage. You should remove the 'For review' link.
- If you are requesting an article processing charge waiver, you must select the relevant waiver option (if requesting a discretionary waiver, the form should have been uploaded at Step 3 'File upload' above).
- If you have uploaded ESM files, please ensure you follow the guidance at <https://royalsociety.org/journals/authors/author-guidelines/#supplementary-material> to include a suitable title and informative caption. An example of appropriate titling and captioning may be found at https://figshare.com/articles/Table_S2_from_Is_there_a_trade-off_between_peak_performance_and_performance_breadth_across_temperatures_for_aerobic_scope_in_teleost_fishes_/3843624.

Author's Response to Decision Letter for (RSOS-202213.R0)

See Appendix C.

Decision letter (RSOS-202213.R1)

Dear Dr Carney,

It is a pleasure to accept your manuscript entitled "Harvesting strategies as evidence for 4000 years of camas (*Camassia quamash*) management in the North American Columbia Plateau" in its current form for publication in Royal Society Open Science.

Due to rapid publication and an extremely tight schedule, if comments are not received, your paper may experience a delay in publication. Royal Society Open Science operates under a continuous publication model. Your article will be published straight into the next open issue and this will be the final version of the paper. As such, it can be cited immediately by other

researchers. As the issue version of your paper will be the only version to be published I would advise you to check your proofs thoroughly as changes cannot be made once the paper is published.

on behalf of Professor Leslie Brown (Associate Editor) and Kevin Padian (Subject Editor)
openscience@royalsociety.org

Appendix A**ROYAL SOCIETY
OPEN SCIENCE****Harvesting strategies as evidence for 4,000 years of camas
(*Camassia quamash*) management in the North American
Columbia Plateau**

Journal:	Royal Society Open Science
Manuscript ID	RSOS-202213
Article Type:	Research
Date Submitted by the Author:	04-Dec-2020
Complete List of Authors:	Carney, Molly; Washington State University, Anthropology Tushingham, Shannon; Washington State University, Department of Anthropology McLaughlin, Tara; Kalispel Tribe of Indians, Kalispel Natural Resources Department d'Alpoim Guedes, Jade; Scripps Research Institute, Department of Anthropology
Subject:	ecology < BIOLOGY, plant science < BIOLOGY
Keywords:	Traditional Resource Management, Geophyte, Pacific Northwest, Harvesting, Foragers
Subject Category:	Organismal and Evolutionary Biology

Author-supplied statements

Relevant information will appear here if provided.

Ethics

Does your article include research that required ethical approval or permits?:

Yes

Statement (if applicable):

The archaeological data herein was excavated in the 1980's; the final report does not contain the permit number for those excavations. We submitted and were approved a research request form with the Tribe's Cultural Resources Department and accessed the curated archaeological material at the Kalispel Tribe's curational facility with full permission.

Data

It is a condition of publication that data, code and materials supporting your paper are made publicly available. Does your paper present new data?:

Yes

Statement (if applicable):

All data used in the above analyses are archived at <http://doi.org/10.5281/zenodo.4287583>. All notes, artifacts, and samples from the project are curated at the Kalispel Tribe's curational facility in Usk, Washington State.

Conflict of interest

I/We declare we have no competing interests

Statement (if applicable):

CUST_STATE_CONFLICT :No data available.

Authors' contributions

This paper has multiple authors and our individual contributions were as below

Statement (if applicable):

M.C. and J.A.d.G. designed the research, M.C. and T.M performed research; M.C., J.A.d.G, S.T., and T.M. analyzed data and wrote the paper.

Harvesting strategies as evidence for 4,000 years of camas (*Camassia quamash*) management in the North American Columbia Plateau

Molly Carney^{1*}, Shannon Tushingham¹, Tara McLaughlin², and Jade d'Alpoim Guedes³

¹Washington State University, Department of Anthropology, College Hall, Pullman, WA 99164, United States

²Department of Natural Resources, Kalispel Tribe of Indians, PO Box 39, Usk, WA 99180, United States

³University of California, San Diego, Department of Anthropology, Scripps Institution of Oceanography, 9500 Gillman Drive, La Jolla, CA 92093, USA, United States

Keywords: Traditional Resource Management, Geophyte, Pacific Northwest, Harvesting, Foragers

1. Summary

One of the greatest archaeological enigmas is in understanding the role of decision-making, intentionality, and interventions in plant life cycles by foraging peoples in transitions to and from low-level food production practices. We bring together archaeological, paleoecological, and botanical data to explore relationships over the past 4,000 years between people and camas (*Camassia quamash*), a perennial geophyte with an edible bulb common across the North American Pacific Northwest. Throughout the mid to late Holocene, people began experimenting with selective harvesting practices through targeting sexually mature bulbs by 3,500 BP, with bulb harvesting practices akin to ethnographic descriptions firmly established by 1,000 BP. While we find no evidence that such interventions lead to a selection for larger bulbs or a reduction in time to maturity, archaeological bulbs do exhibit several other domestication syndrome traits. This establishes considerable continuity to human intervention into camas life cycles, but these dynamic relationships did not result in unequivocal morphological indications of domestication. This approach to tracking forager plant management practices offers an alternative explanatory framework to conventional management studies, supplements oral histories of Indigenous traditional resource management, and can be applied to other vegetatively propagated species.

2. Introduction

For almost all human groups through time and space, geophyte plants offer important and reliable sources of carbohydrates, nutrients, and economic products. Geophytes, with their edible underground storage organs, are hypothesized to have been of critical importance to early hominin development (1) and use by *Homo sapiens* has been documented as early as 170,000 thousand years ago(2). Despite this, the complex geographical and historical pathways of managed and domesticated geophytes are still poorly understood(3). Within North America, establishing timelines and trajectories of geophyte plant use and cultivation by hunter-gatherer groups has been particularly challenging (4, 5), notwithstanding strong ethnohistoric evidence for their place as staple food sources in many Indigenous economies.

*Molly Carney (molly.carney@wsu.edu)

†Present address: Washington State University, Department of Anthropology, College Hall, Pullman, WA 99164, United States

Understanding geophyte management practices in foraging societies and when these entanglements between climate, environment and food resources become deeply embedded in sociocultural systems is crucial to our collective knowledge of what happens when management does not lead to selection or domestication(6), and there are few studies which focus on these alternative pathways. We see several reasons for this. Most paleobotanical research examines human exploitation of plants, primarily through the lens of domestication and selection, rather than through reciprocal interactions between humans and plants (7-9). The domestication process itself is viewed as the result of phenotypic and genetic changes in cultivars that are different from unmanaged populations (10), a process made even more difficult with the low chances of geophyte preservation in the archaeological record. It is now commonly accepted that these processes are co-evolutionary, multi-directional pathways or entanglements between human and plant or animal species (11-13), and nonhuman species may also have a role in selection for domestication syndrome traits (14). As the centers of domesticate origins increase in number and are continuously refined (10, 15), it has become evident that varying ecologies, niches, aspects of plant physiology, and cultural traditions have resulted in many different pathways to domestication and domestication syndromes. Indeed, the phenotypic and genetic changes we see today in contemporary domesticated plant species are likely the end results of various unconscious decisions and behaviors occurring over millennia (16).

In western North America, it is widely recognized that Indigenous hunting and gathering communities managed plants through a variety of practices and cultural institutions (17, 18). Signatures of anthropogenic burning (19), seasonal harvesting (20), archaeological recognition of garden features (21), and identifying plant species range extensions (4, 22) have been used to document past plant management practices in western North America. And while ethnographic work and oral histories attest to the presence of intimate human-plant symbiotic relationships for many plant food species, until now studies have not produced corresponding direct empirical evidence in the form of plant remains for significant archaeological geophyte management practices. Below we report archaeological evidence for substantial management of a cultural keystone geophyte species(24) and demonstrate considerable time depth of ethnohistoric plant stewardship practices which do not lead to domestication pathways.

Figure 1. Relief map of the Pend Oreille Valley, northeastern Washington State, with location of valley outlined in light red.

Camassia quamash (Pursh) Greene, commonly known as camas, is a flowering herbaceous perennial with an edible bulb that is considered among many Northwest Indigenous communities to be the most important food source (25). Many have hypothesized that past Indigenous management practices positively impacted camas growth and production and that the plant may have been semidomesticated (26). In fact, over 100 years ago renowned botanist Luther Burbank's experiments illustrated that traits within camas species (*Camassia* spp.) are easily selected for and produced notable increases in bulb sizes after only two generations (27). Archaeological and genetic studies, however, have not yet produced evidence for human-mediated change within the plants genotype or phenotype (28, 29). Here we present archaeological evidence from the Pend Oreille Valley, in northeastern Washington State (Fig. 1), for two distinct camas harvesting patterns, one of which reflects conscious human intervention into the plant's life cycle. We show that harvesting strategies changed throughout our 4,000 year study period, and further illustrate that such human-plant interactions resulted in a stable yet dynamic subsistence system in which people and camas coexisted in a symbiotic relationship that did not result in dependence on humans for reproduction or other phenotypic change.

3. Human Relationships with Camas Life Cycles

The North American genus *Camassia*, taxonomically classified to the Asparagaceae family, is comprised of five species with ranges across the United States and southern Canada (30). *Camassia quamash* (common camas) is the most widespread species across the Pacific Northwest. Common camas is a bulbous perennial with inflorescences of large, blue flowers which prefers to live in colonies in poorly drained fields or prairies with xeric moisture regimes (Fig. 2). The bulbs are perennating underground storage organs that consist of a shortened stem and leaf bases, as well as a basal plate from which fibrous roots emerge. These bulbs contain high amounts of inulin, and are usually prepared by steaming or roasting in earth ovens for long periods of time (24-72 hours) at high temperatures to convert the complex carbohydrates into easily digestible fructose (31). Both substantial soil moisture from winter through spring and unrestricted drainage are necessary for optimal growth (32). Excessive summer soil moisture or extreme heat will reduce bulb yields, while cold temperatures do not appear to impact productivity. Moreover, common camas needs 42-100 days of cold temperatures (<5 °C) to germinate (33).

Figure 2. Common camas bulbs illustrating main features in the text. (A) Camas in bloom. (B) Example of whole bulbs dated to 1766 BP. (C) Scanning electron micrograph of a bulb cross-section showing heat-fused leaf scales. (D) Scanning electron micrograph of common camas epidermis.

Ethnographic evidence indicates that camas was intensely managed and highly valued throughout the Northwest (25, 34). Contemporary practice and historic records indicate that camas was harvested with wooden digging sticks during the summer months after flowers had faded but stalks and seed capsules were still visible (25). During bulb harvests seed capsules and dried stalks were replanted along with any uprooted immature bulbs. Fields were also intentionally burned in some places. Such practices not only aerated soils but kept fields clear, removed competitors, and enhanced plot productivity (26). It is unknown when these practices became established, or how they may have changed throughout time. Previous research has established that camas has been consumed for over 8,000 years (28), was traded (34) and had peaks and nadirs in consumption (35). Camas production and storage is widely assumed to have been intensified by four to three thousand years ago within the Columbia Plateau (35, 36).

Several *Camassia* spp. life history traits, including bulb anatomy and time to maturity, are important to understanding human management. Camas plants reproduce primarily by seed, but it is estimated that 15% of sexually mature bulbs will reproduce asexually through offset bulblets (26, 35). Seedlings take between 4-5 years to reach sexual maturity (33). Bulbs have interior layered leaf scales and consist of two parts: a mother bulb and its enclosed daughter bulb. In late winter or early spring new leaves are developed as part of the daughter bulb. These tissues are completely replaced annually (26). While camas bulbs do not increase in size at a linear rate (26), leaf scales grow in regular layers and increase in number over time. Sexually mature bulbs of 4-5 years or greater in age will have 3-7 leaf scales while immature bulbs of three years or less often have 1-3 leaf scales (33, 35) (Fig. 2c).

Based on this understanding of camas life cycles, previous camas growth experiments, ethnohistoric sources and contemporary traditional camas knowledge we predict that past peoples engaged in strategic harvesting practices that involved harvesting only the sexually mature bulbs, replanting immature bulbs for sustainable fields and future harvests (Table 1). We further hypothesize that human selection may have resulted in a reduction in time to maturity and/or increase in bulb size. To evaluate these hypotheses, we compared the leaf scales of archaeological camas bulbs from five riverine sites in the Pend Oreille Valley of northeastern Washington State, United States, with those from a comparative collection of modern, experimentally grown mature and immature camas bulbs. Below, we report these data along with 24 existing and 2 new radiocarbon dates (Supplemental Dataset 1, Supplemental Figure 1) to evaluate bulb harvests across time and climatic conditions.

Table 1. Camas harvesting strategies, bulb maturities, archaeological evidence, and human labor and resource conservation considerations.

Strategy	Life cycle stage at harvest	Archaeological evidence	Implications
“stripping”	Bulbs across all ages/maturity collected	Wide range of leaf scales and bulb sizes or larger proportion of immature bulbs (≤ 3 leaves)	Lower associated time costs, may deplete resources locally
“selective harvesting”	Predominantly sexually mature camas bulbs, assume immature plants are replanted	Primarily mature leaf scale numbers (4+ leaves)	Greater time inputs at harvest, but actions ensure future harvests

4. Methods

4.1 Archaeological Materials. The materials analyzed in this study include 110 carbonized *Camassia quamash* bulbs from five sites along the Pend Oreille River in Kalispel ancestral lands, northeastern Washington State (Fig. 1). Ethnographic, archaeological, and historical ecological data from the Pend Oreille Valley, WA indicate this valley was among the most renowned camas harvesting grounds among interior northwest groups (37, 38). Most of the analyzed bulbs were part of legacy collections associated with four sites salvage excavated during the Calispell Valley Archaeological Project (CVAP) (39). Bulbs from a fifth site (45PO435) were recovered in 2014-2015 field seasons excavations conducted by Eastern Washington University and Washington State University (40). The bulbs selected for analysis were recovered from well-provenienced archaeological features with closely associated radiocarbon dates including earth ovens, hearths, house floors,

and one special-use structure (Supplemental Dataset 1). All radiocarbon dates were re-calibrated using OxCal 4.4.1 with the IntCal 20 curve (SI Fig. 1); median calibrated dates are used in the rest of this analysis (41, 42).

4.2 Paleoethnobotanical Methods. We examined over 100 modern, nursery grown *Camassia quamash* bulbs of known ages for comparative data on common camas lifecycles and ages (Supplemental Dataset 1) (43). Fifty specimens each of mature bulbs (>4 years of age) and immature bulbs (2-3 years of age) were measured, including bulb length, width, and thickness to approximate bulb volume as well as bulb weight. Bulbs were measured fresh shortly after harvesting during summer. They were then wrapped in aluminum foil and placed within sand to create an anaerobic environment similar to earth ovens, and experimentally charred in a muffle furnace at 350° C for one hour. Bulbs were re-measured, and thirty of each group of modern bulbs were then dissected and number of leaf scales counted.

Archaeological, carbonized bulbs were identified as *Camassia quamash* by Stenholm (2000) and confirmed by the lead author (Supplemental Dataset 2). Archaeological bulb length, width, and thickness, an estimate of bulb preservation, and number of visible leaf scales for each of the carbonized bulbs were recorded. Bulb length, width, and thickness were recorded to the .01 mm using a Neiko stainless steel digital caliper. Weight was recorded to the .001 g using a digital scale. Bulb preservation was estimated based upon fragmentation (i.e. whole bulbs without any breakage were deemed 100% preserved). When bulbs included some breakage (i.e., less than 100% preserved), it was sometimes possible to examine the basal plate or broken emergent scape to record the number of leaf scales present. We randomly subsampled 20% of all bulbs, dissected a cross-section, and counted number of leaf scales.

We also calculated a simple ratio to approximate bulb size. Camas bulbs vary between “pencil-like” to teardrop shaped to spherical (26), making bulb volume calculations difficult. It is also estimated that 40% of bulb weight is lost during steaming preparation and an additional 25% lost during charring, though general bulb shape is preserved (44). Given these taphonomic considerations, we employ a ratio of bulb diameter to bulb length to establish a proxy for bulb size.

$$\frac{\text{width} + \text{thickness}}{2} \\ \text{length}$$

All archaeological bulbs with leaf scale data and associated with radiocarbon dated features were binned into groups by 500-year increments. Leaf scale data were graphically plotted to assess similarity and differences in archaeological and modern camas samples. These relationships were tested in R 4.0.2 using the Mann-Whitney U test with the Benjamini and Hochberg (45) p-value adjustment for multiple comparisons and the Kruskal-Wallis rank sum test, a non-parametric equivalent to ANOVA. All statistical significance inferences are made via comparison with a 0.05 confidence level and calculated effect sizes.

5. Results

5.1 Proxies for maturity. For the modern comparative camas bulbs, our results confirm previous findings (26, 28) that bulb dimensions varied widely and were not adequate proxies for bulb maturity (charred size ratio: $U=113.5$, $p=1.128 \times 10^{-6}$). The quantity of interior bulb leaf scales, however, were significantly different between mature and immature samples ($p=1.186 \times 10^{-9}$, $U=60$). Based upon these comparative data we conclude that bulb leaf scales are an excellent approximation of relative bulb age, as sexually immature bulbs have fewer leaves (3 or less) and sexually mature bulbs have more leaves (4-5 or more).

5.2 Camas harvesting strategies. People alternated between two camas harvesting strategies throughout much of the study period (Fig. 3). Archaeological leaf scale data for the oldest temporal period, 3999-3500 BP, have a large range. The Mann-Whitney U test indicates bulbs are different than both mature bulb populations ($U=148$, $p=0.01038$, $\eta^2=0.181$) but with size ratios that are comparable to the size ratios of other periods (Figure 4c, see below). We see this period as a time in which people opportunistically collected camas bulbs regardless of maturity (“stripping”).

A second harvesting strategy is evident across several temporal periods including the 3499-3000 BP period, 1999-1500, and 999-present intervals. Comparisons between the archaeological sample distributions are different than the immature control data, with strong effect sizes ($\eta^2 > 0.2$) and are statistically indistinguishable at the 0.05 confidence level from the mature control population (Supplemental Table 1). These data indicate that most camas bulbs harvested during these periods were sexually mature bulbs; we infer that most, if not all, immature bulbs were replanted or not harvested.

Figure 3. Comparisons of camas bulb leaf scale data by temporal period.

The data for 2499-1000 BP indicate oscillations between harvesting practices. For both the 2499-2000 and 1499-1000 BP temporal periods, archaeological leaf scale data indicate that people were much more likely to harvest bulbs across all maturity levels (mean leaf scales of 2.875 and 2.6538 respectively). Archaeological sample distributions are significantly different than mature camas bulb populations at the 0.05 confidence level (Supplemental Table 1 and large effect size values ($\eta^2 > 0.18$)). To contrast this, the 1999-1500 temporal period contains bulbs from a single feature that average 3.5 leaf scales. For this period comparisons between the archaeological bulb sample distributions are different than the immature control data ($p=0.01056$, $\eta^2=0.23$), and indicate harvesting of older bulbs.

5.3 Evidence for management. To assess for human selection within carbonized bulb storage organs, we also tested ratio and weight variables to assess change in bulb size relative to bulb maturity. The Kruskal-Wallis test indicates that bulb size does not vary significantly throughout time ($\chi^2=9.8061$, $p=0.2789$, $\epsilon^2=0.0875$), nor does bulb weight ($\chi^2=7.5728$, $p=0.376$, $\epsilon^2=0.223$). We conclude that there is no evidence to support our second hypothesis that Pend Oreille Valley camas bulbs were selected for an increase in size or weight, nor for a reduction in time to maturity. Furthermore, as bulb size and weight do not vary significantly through time, we see no evidence for climatic influences on camas bulb size.

We further compared modern and archaeological camas traits to domestication syndrome traits in asexually propagated geophytes (Supplemental Table 1) (3). There are seven possible vegetative crop domestication syndrome traits for *Camassia* spp. We see supporting evidence for three traits: camas is easily harvested, can be harvested across multiple seasons, with about 15% reproducing asexually through offshoot bulblets (35). While one study found no evidence for human-mediated translocation across a small portion of the Northwest region, ethnohistoric records and contemporary restoration efforts indicate camas bulbs are easily transplanted and propagated (26, 46). Given this lack of evidence for an increase in size over time, size standardization, or decrease in time to maturity, we infer that these traits are the result of long-term management rather than conscious or unconscious selection.

Table 2. Presence and absence of vegetative crop domestication syndrome traits within *Camassia* spp. after Denham et al. 2020. Denham et al. (2020:9) consider bulbous underground storage organs as primarily propagated vegetatively and here we follow their criteria for domestication in asexually propagated plants.

Trait category	Domestication in asexually propagated plants	Modern camas traits	Archaeological camas trait presence
Mode of reproduction	1. Partial or complete loss of sexual reproduction ability 2. Increased uniformity in clonal reproduction traits	1. Estimated 15% reproduce asexually (35) 2. Unknown	1. Unknown 2. Unknown
Plant life cycle	Shift towards biennial to annual life cycle	Perennial life cycle	Perennial life cycle; no evidence for decrease in time to maturity (this study)
Yield of edible portion	Increased size	No evidence (26) Potential to increase size over several generations (27)	No evidence for change in size over time (⁷ , this study)
Ease of harvesting	Development of easily separated U.S.C.s	Easily harvested with digging sticks; many bulbs are easily separated (47)	Unknown, but presence in many archaeological sites indicates easily harvested
Timing of production	Asynchronous and more continuous production, with in-ground storage	Usually harvested in summer, but can be harvested in winter as well (47)	Unknown; ethnohistoric evidence of multi-season harvests; potential to look at bulb phenol (20)
Environmental tolerance	Traits that enable cultivation across wider environmental range	Popular species among gardeners and bulb enthusiasts Easily transplanted in restoration contexts	No evidence for human dispersal (29) Archaeological presence beyond modern range (34) interpreted as trade Historically recorded beyond contemporary range (4)

6. Discussion

Climate and camas consumption. By at least 4,000 years ago, camas was an integral part of the diet for residents of the Columbia Plateau and was processed on a large scale within the Pend Oreille Valley (35). The earliest dated material in our study corresponds with a climatic period of pronounced seasonality (i.e. cooler, wetter winters than present day) (48). As camas plants thrive with cooler, dormant periods and warm and dry summer periods, it is possible that this temporal period is correlated with an abundance of camas and expansion of productive habitat. We interpret this period as one in which social institutions describing how these plants were harvested were not yet established, or that such cultural rules were not strictly followed during a period of possible abundance. Between 4000-3,500 cal BP, practices of bulk-processing camas became fully routine, and camas was established as a staple overwintering food (35).

In the subsequent temporal period, it appears that residents and visitors to the Pend Oreille Valley adopted a new camas harvesting strategy. Primarily mature bulbs were collected and cooked from 3,500-2,500 cal BP (“selective harvesting”). Thus, by ~3,500 cal BP people were consciously and deliberately only harvesting the bulbs of sexually mature plants, and we infer replanting immature bulbs and seed capsules. These selective harvesting practices exhibit significant care for camas plants and fields, such that these established human-camas relationships productively benefited both plant and human species. Bulbs in this period were also the largest (Fig. 4c). Climatically, from 3,500-2000 cal BP there is a sharp cooling event with increased cold season precipitation evident in $\delta^{18}O$ and sea surface temperature reconstructions (Fig. 4 a, b) (48). Camas plants

readily tolerate cold temperatures and thrive with winter precipitation. This globally recognized cold period (49) then should not have negatively affected camas niches.

By 2,500 cal BP, however, these relationships changed. Our results indicate that people shifted from a focus on harvesting mature plants to once again harvesting bulbs across maturity levels. In the bulb data from the 2,500-2,000 and 1,500-1,000 cal BP periods, leaf scale data is significantly different than that of mature bulbs. We suggest that this is evidence for either a relaxation of the social rules governing camas harvests, or the influence of other, unidentified factors influencing human-plant relationships and access to camas fields. At 2,200 cal BP the northern Columbia Plateau climate begins to gradually transition from wetter to drier conditions, and there follows a 500-year period which primarily mature bulbs were collected. These data come from a single earth oven feature, and we argue that more data from archaeological camas bulbs are necessary to understand these oscillating harvesting strategies.

Incidentally, this 2,500-1,000 cal BP period corresponds with widespread changes throughout the Columbia-Fraser Plateau culture area. Intermittently throughout the Columbia-Fraser Plateau, people begin to abandon semi-sedentary lifestyles for more mobile patterns of movement. In many places, pithouses, the architectural residence of choice for 3000 years prior, are abandoned in favor of lightweight mat lodges (50-52). Rock shelters were re-purposed from habitation sites to storage caches (53). Trade and human movement increase throughout the region (54), and on the coast, there is evidence for the trade of camas across hundreds of kilometers (34). These significant material culture changes most likely reflect other dramatic changes in everyday life. From 2,000-1,000 cal BP precipitation alternates along smaller decadal to centennial oscillations and are overall less varied when compared to the temporal periods before and after. Further climatic resolution is needed to understand these correlations as well as the causes and effects of such dramatic lifestyle shifts during this period.

By 1,000 cal BP people return to primarily harvesting mature bulbs. This selective harvesting pattern continues through to the contact period, and is likely the pattern from which our ethnographic sources for plant management and stewardship arise. Despite global and regional evidence for the Little Ice Age cooling event, archaeological activity does not subside during this period for the valley (39), nor are there significant changes in bulb maturity, size, or weight at harvest. We suggest that the cultural institutions surrounding camas harvests ensured ongoing, sustainable harvests despite a subtle decrease in regional precipitation throughout the late Holocene (48, 55) and global cooling events such as the Little Ice Age (56).

Figure 4. Summary of Pend Oreille Valley human-camas relationships through time. (A) $\delta^{18}\text{O}$ data from Cleland lake, British Columbia (48). (B) Northeastern Pacific sea surface temperature ($^{\circ}\text{C}$) reconstruction⁵⁸. (C) Median size ratio for bulbs by temporal period. (D) Median number of leaves by temporal period.

Management without selection. While many who work closely with Indigenous communities suggest that camas may have been domesticated in some form, our research supports other studies that do not find unequivocal evidence for selection or human-aided dispersal patterns (28, 29). Contemporary domestication

1
2 explanations see the repetitive cycle of sowing, collecting, and sowing of wild plants as the primary processes
3 through which novel phenotypic and genotypic traits emerge(58), yet here we have a case in which such
4 human behaviors and intimate knowledge of plant lifecycles do not act as a selective forces. These interactions
5 should not be seen as incipient agriculture, but instead flexible relationships with plants that resulted in
6 dynamic yet stable economic systems (8).
7

8
9 There is increasing evidence that across the Americas, societies consistently maintained mixed subsistence
10 strategies for millennia, relying partially on the cultivation of domesticated plants or management of non-
11 domesticates, and that many of these practices do not result in large-scale food production (8, 59). Leading
12 scholars (5, 8, 15) now argue that agriculture is not an inevitable product of domestication. We further add
13 that domestication is not an automatic outcome of management or extensive niche construction activities(6).
14 As scholarly dialogue continues to examine the roles and influences plants have upon people and the
15 entangled relationships that ensue (9, 11), studies which prioritize human-plant relational interactions offer
16 fresh frameworks for theorizing and examining alternative modes of production and the development of
17 stable yet dynamic non-agricultural subsistence systems.
18
19

20
21 **Traditional ecological knowledge informs the past and the future.** This study establishes patterns of camas
22 harvesting strategies and behaviors for one valley within the North American Columbia Plateau from 4,000
23 years ago through the post-contact period. Several results support ethnographic descriptions of camas
24 harvests throughout the northwest and supplant our knowledge of how such behaviors may have changed
25 over time. These results also support Indigenous oral traditions and contemporary practices of selective
26 harvesting and plant food resource stewardship, extending traditional ecological knowledge systems at least
27 3,500 years back into time. We may also use these insights into human-camas relationships to model
28 contemporary camas management and harvests, ensuring sustainable food options for Indigenous
29 communities working to restore and reclaim autonomy over their health, well-being, and cultural heritage.
30

31 We also demonstrate that for camas and other perennial geophyte plants, maturity at harvest may be a more
32 fruitful variable than size in researching questions of cultivation, domestication, or management. We suggest
33 that others investigating human-geophyte interactions and geophyte domestication pathways consider plant
34 life history traits and time to sexual or asexual maturity in future studies. This approach offers an alternative
35 explanatory framework to conventional management studies and can be applied to other vegetatively
36 propagated species. Such a perspective provides significant potential across research agendas spanning
37 forager-plant relationships through time, the advent of cultivation practices, teasing out varied domestication
38 pathways, and may even contribute to hominin geophyte use and human evolutionary development.
39
40

41 **Acknowledgments**

42 We are especially grateful to the Kalispel Tribe for allowing us to work with and understand the history of their ancestors.
43 We also wish to thank Stephenie Kramer for inspiring this study, and Cassidy Fairlane and Clare Carney for help with
44 Figure 1.
45

46 **Ethical Statement**

47 Research on humans must include a statement detailing ethical approval and informed consent. Research using animals
48 must adhere to local guidelines and state that appropriate ethical approval and licences were obtained. Please read our
49 editorial policies carefully before submission.
50

51 **Funding Statement**

52 This work was supported by the Association for Washington Archaeology Student Travel Grant, the Washington State
53 University Graduate and Professional Student Association Dissertation Grant, the Washington State University College of
54 Arts and Sciences Boeing Fellowship in Environmental Studies, and the University of California, San Diego.
55

56 **Data Accessibility**

57 All data used in the above analyses are archived at <http://doi.org/10.5281/zenodo.4287583>. All notes, artifacts, and samples
58 from the project are curated at the Kalispel Tribe's curational facility in Usk, Washington State.
59

60 **Competing Interests**

The authors declare no competing interests.

Authors' Contributions

M.C. and J.A.d.G. designed the research, M.C. and T.M performed research; M.C., J.A.d.G, S.T., and T.M. analyzed data and wrote the paper.

References

1. Hardy K, Brand-Miller J, Brown KD, Thomas MG, Copeland L. The importance of dietary carbohydrate in human evolution. *The Quarterly Review of Biology*. 2015;90(3):251-68.
2. Wadley L, Backwell L, d'Errico F, Sievers C. Cooked starchy rhizomes in Africa 170 thousand years ago. *Science*. 2020;367(6473):87-91.
3. Denham T, Barton H, Castillo C, Crowther A, Dotte-Sarout E, Florin SA, et al. The domestication syndrome in vegetatively propagated field crops. *Annals of Botany*. 2020;125(4):581-97.
4. Lepofsky D, Lertzman K. Documenting ancient plant management in the northwest of North America. *Botany*. 2008;86(2):129-45.
5. Smith BD. Niche construction and the behavioral context of plant and animal domestication. *Evolutionary Anthropology: Issues, News, and Reviews*. 2007;16(5):188-99.
6. Mercuri AM, Fornaciari R, Gallinaro M, Vanin S, di Lernia S. Plant behaviour from human imprints and the cultivation of wild cereals in Holocene Sahara. *Nature Plants*. 2018;4(2):71-81.
7. Lodwick LA. Agendas for Archaeobotany in the 21st Century: data, dissemination and new directions. *Internet Archaeology*. 2019;53.
8. Fausto C, Neves EG. Was there ever a Neolithic in the Neotropics? Plant familiarisation and biodiversity in the Amazon. *Antiquity*. 2018;92(366):1604-18.
9. van der Veen M. The materiality of plants: plant–people entanglements. *World Archaeology*. 2014;46(5):799-812.
10. Fuller DQ, Denham T, Arroyo-Kalin M, Lucas L, Stevens CJ, Qin Li, et al. Convergent evolution and parallelism in plant domestication revealed by an expanding archaeological record. *Proceedings of the National Academy of Science, USA*. 2014;111(17):6147-52.
11. Fuller DQ, Stevens C, Lucas L, Murphy C, Qin L. Entanglements and entrapments on the pathway toward domestication. In: Der L, Fernandini F, editors. *The Archaeology of Entanglement*. Walnut Creek, CA: Routledge; 2016. p. 151-72.
12. Rindos D. *The origins of agriculture: an evolutionary perspective*. London: Academic Press, Inc.; 1984.
13. Zeder MA. The domestication of animals. *Journal of anthropological research*. 2007;68(2):161-90.
14. Spengler RN. Origins of the Apple: The Role of Megafaunal Mutualism in the Domestication of Malus and Rosaceous Trees. *Front Plant Sci*. 2019;10:617.
15. Zeder MA. Core questions in domestication research. *PNAS*. 2015;112(11):3191-8.
16. Fuller DQ, Asouti E, Purugganan MD. Cultivation as slow evolutionary entanglement: comparative data on rate and sequence of domestication. *Vegetation History and Archaeobotany*. 2012;21(2):131-45.
17. Anderson K. *Tending the wild: Native American knowledge and the management of California's natural resources*. Berkeley, California: University of California Press; 2005.
18. Deur D, Turner NJ, editors. *Keeping it living: traditions of plant use and cultivation on the Northwest Coast of North America*. Seattle, WA: University of Washington Press; 2005.
19. Lightfoot KG, Cuthrell RQ, Striplen CJ, Hylkema MG. Rethinking the study of landscape management practices among hunter-gatherers in North America. *American Antiquity*. 2013;78(2):285-301.
20. Gill KM. Seasons of Change: Using Seasonal Morphological Changes in Brodiaea Corms to Determine Season of Harvest from Archaeobotanical Remains. *American Antiquity*. 2014;79(4):638-54.
21. Hoffman T, Lyons N, Miller D, Diaz A, Homan A, Huddleston S, et al. Engineered feature used to enhance gardening at a 3800-year-old site on the Pacific Northwest Coast. *Science Advances*. 2016;2(12).

22. Ritland K, Meagher LD, Edwards DGW, El-Kassaby YA. Isozyme variation and the conservation genetics of Garry oak. *Canadian Journal of Botany*. 2005;83(11):1478-87.
23. Armstrong CG, Dixon WcM, Turner NJ. Management and Traditional Production of Beaked Hazelnut (k'áp'xw-az', *Corylus cornuta*; Betulaceae) in British Columbia. *Human Ecology*. 2018;46(4):547-59.
24. Garibaldi A, Turner N. Cultural Keystone Species: Implications for Ecological Conservation and Restoration. *Ecology and Society*. 2004;9(3).
25. Turner NJ, Kuhnlein HV. Camas (*Camassia* spp.) and riceroot (*Fritillaria* spp.): two liliaceous “root” foods of the Northwest Coast Indians. *Ecology of Food and Nutrition*. 1983;13(4):199-219.
26. Beckwith BR. The queen root of this clime: ethnoecological investigations of blue camas (*Camassia leichtlinii* (Baker) Wats., *C. quamash* (Pursh) Greene; Liliaceae) and its landscapes on southern Vancouver Island, British Columbia. Victoria, BC: University of Victoria; 2004.
27. Burbank L. The *Camassia* – will it supplant the potato? In: Whitson RJ, William HS, editors. *Luther Burbank: His Methods and Discoveries and Their Practical Application*. 71914.
28. Kramer S. *Camas Bulbs, the Kalapuya, and Gender: Exploring Evidence of Plant Food Intensification in the Willamette Valley of Oregon*. Eugene, OR: University of Oregon; 2000.
29. Tomimatsu H, Kephart SR, Vellend M. Phylogeography of *Camassia quamash* in western North America: postglacial colonization and transport by indigenous peoples. *Mol Ecol*. 2009;18(18):3918-28.
30. Gould FW. A systematic treatment of the genus *Camassia* Lindl. *The American Midland Naturalist*. 1942;28(3):712-42.
31. Konlande JE, Robson JRK. The nutritive value of cooked camas as consumed by Flathead Indians. *Ecology of Food and Nutrition*. 1972;2:193-5.
32. Statham DS. A Biogeographic Model of Camas and Its Role in the Aboriginal Economy of th Northern Shoshoni in Idaho. *Tebiwa*. 1975;18(1):59-80.
33. Leffingwell AM. Morphological study of bulb and flowers of *Camassia quamash* (Pursh) Greene. *Research studies of the State College of Washington*. 1930;2:80-9.
34. Lyons N, Ritchie M. The Archaeology of Camas Production and Exchange on the Northwest Coast: With Evidence from a Sts'ailes (Chehalis) Village on the Harrison River, British Columbia. *Journal of Ethnobiology*. 2017;37(2):346-67.
35. Thoms AV. *The northern roots of hunter-gatherer intensification: camas and the Pacific Northwest*. Pullman, WA: Washington State University; 1989.
36. Fulkerson TJ, Tushingham S. ??? in press.
37. Anastasio A. The Southern Plateau: An Ecological Analysis of Intergroup Relations. *Northwest Anthropological Resource Notes*. 1972;6(2):109-229.
38. Stenholm NA. Botanical Analysis for the Calispell Valley Archaeology Project. In: William Andrefsky J, Burtchard GC, Presler KM, Samuels SR, Sanders PH, Thoms A, editors. *The Calispell Valley Archaeological Project Final Report*. Pullman, WA: Project Report No. 16, Center for Northwest Anthropology; 2000. p. 14.1-66.
39. Andrefsky Jr. W, Burtchard GC, Presler KM, Samuels SR, Sanders PH, Thoms A, editors. *The Calispell Valley Archaeological Project Final Report*. Pullman, Washington: Center for Northwest Anthropology; 2000.
40. Dorwin JT. Box Canyon Hydroelectric Project Archaeological Evaluation of 45PO174 and 45PO435. Usk, WA: Kalispel Tribe of Indians; 2018.
41. Bronk Ramsey C. Bayesian analysis of radiocarbon dates. *Radiocarbon*. 2009;51(1):337-60.
42. Reimer PJ, Austin WEN, Bard E, Bayliss A, Blackwell PG, Bronk Ramsey C, et al. The IntCal20 Northern Hemisphere radiocarbon age calibration curve (0–55 cal kBP). *Radiocarbon*. 2020;62:1-33.

43. Carney M, d'Alpoim Guedes J. Paleoethnobotanical identification criteria for bulbs of the North American Northwest. *Vegetation History and Archaeobotany*. in press.
44. Stenholm NA. Botanical Analysis of Floral Samples. In: Ames KM, Cornett WL, Hamilton SC, editors. *Archaeological Investigations (1991-1995) at 45CL1 (Cathlapotle): Clark County Washington: A Preliminary Report*. Portland, OR: Wapato Archaeology Project Report 6. Department of Anthropology, Portland State University; 1996.
45. Benjamini Y, Hochberg Y. Controlling the false discovery rate: a practical and powerful approach to multiple testing. *Journal of the Royal Statistical Society Series B*. 1995;57:289-300.
46. Stanley AG, Dunwiddie PW, Kaye TN. Restoring Invaded Pacific Northwest Prairies: Management Recommendations from a Region-Wide Experiment. *Northwest Science*. 2011;85(2):233-46.
47. Turner NJ. *Ancient Pathways, Ancestral Knowledge: Ethnobotany and Ecological Wisdom of Indigenous Peoples of Northwestern North America*. Montreal, Quebec: McGill-Queen's University Press; 2014.
48. Steinman BA, Pompeani DP, Abbott MB, Ortiz JD, Stansell ND, Finkenbinder MS, et al. Oxygen isotope records of Holocene climate variability in the Pacific Northwest. *Quaternary Science Reviews*. 2016;142:40-60.
49. Marcott SA, Shakun JD, Clark PU, Mix AC. A reconstruction of regional and global temperature for the past 11,300 years. *science*. 2013;339(6124):1198-201.
50. Ames KM. Sedentism: A Temporal Shift or a Transitional Change in Hunter-Gatherer Mobility Patterns? In: Gregg SA, editor. *Between Bands and States*. 9. Carbondale: Southern Illinois University Press; 1991. p. 108-34.
51. Ames KM. Radiocarbon Dates and Community Mobility Patterns on the Columbia Plateau. *Journal of Northwest Anthropology*. 2012;7:167-94.
52. Chatters JC. Population growth, climatic cooling, and the development of collector strategies on the Southern Plateau, Western North America. *Journal of World Prehistory*. 1995;9(3):341-400.
53. Endacott N. Ages of the Squirt Cave Storage Pits. *Archaeology in Washington*. 1992;IV:39-44.
54. Galm JR. Prehistoric trade and exchange in the Interior Plateau of northwestern North America. In: Baugh TG, Ericson J, E., editors. *Prehistoric Exchange Systems in North America*. New York: Springer; 1994. p. 275-305.
55. Nelson DB, Abbott MB, Steinman B, Polissar PJ, Stansell ND, Ortiz JD, et al. Drought variability in the Pacific Northwest from a 6,000-yr lake sediment record. *Proc Natl Acad Sci U S A*. 2011;108(10):3870-5.
56. Grove JM. *The Little Ice Age*. London: Methuen; 1988.
57. Kienast SS, McKay JL. Sea surface temperatures in the subarctic northeast Pacific reflect millennial-scale climate oscillations during the last 16 kyrs. *Geophysical Research Letters*. 2001;28(8):1563-6.
58. Weis E, Kislev ME, Hartmann A. Autonomous Cultivation Before Domestication. *Science*. 2006;312(5780):1608-10.
59. Killion TW. Nonagricultural Cultivation and Social Complexity. *Current Anthropology*. 2013;54(5):569-606.

Appendix B

Review for Royal Society Open Science (RSOS-202213)

Carney, M., Tushingham, S., McLaughlin, T., d'Alpoim Guedes, J., Harvesting strategies as evidence for 4000 years of camas (*Camassia quamash*) management in the North American Columbia Plateau.

This paper is tracking forager plant management practices. Research focus on harvesting practices by stripping versus targeting sexually mature camas bulbs, through the last 3500 years in the Pacific Northwest, USA. There is strong ethnohistoric evidence camas was a staple food in many Indigenous economies. Carbonized archaeological bulbs are examined for domestication syndrome traits.

Specific comments:

-Fig. 1: 'Kilometers' should read 'kilometer (km)'

-3/21: I guess the authors used old literature ? for the family name 'Asparagaceae' (Gould 1942). Following Mabberly (1993 and later editions) *Camassia* belongs to the Liliaceae.

[Mabberly, D.J., 1993, *The Plant-book*. Cambridge University Press, UK.]

-5/40: explain ANOVA and give a reference.

-Fig. 2: I guess 'metric' suggests the units are 'cm'? Not clearly stated.

-All over manuscript: while in the references years (e.g. 2014) are written without a comma, in the text years are written with/without a comma (e.g. 3,500 BP vs. 1766 BP). Better to eliminate this inconsistency, and, commas in numbers up to 10,000 are unnecessary and detract from readability.

-Table 2 is not introduced in the main text.

-Table 2: the acronym USO is not explained. To serve the reader, better not using acronyms here.

-Figure 4: D18O = $\delta^{18}O$

panels B and C are lacking a unit.

'Age BP' should read 'Age (yr BP)'

-9/28 and elsewhere: in the text "cal BP" and "BP" is mixed up. Does this difference in units express a difference in precision? Please explain in the Method section the units of time.

-The literature used (55, 56, 57) is outdated. I doubt if a comparison with sea surface temperature from the Eastern Pacific makes sense as climate conditions may vary regionally. The "subtle decrease in regional precipitation", the "global cooling events such as the Little Ice Age", and panels 'C' and 'D' are not more than hand waiving arguments. Fig. 4 shows little information and can be deleted.

-I am missing a 'Conclusions' section clearly stating which incremental step forwards in understanding has been made. The current text in "Traditional ecological knowledge informs the past and the future" may be re-structured into a 'Conclusions' section.

-References show some inconsistencies (e.g. PNAS in lines 11/44 vs. 13/39) (science = Science in 13/22)

-References: the abundant use of caps in titles is unnecessary (and inconsistently used).

I have not received Suppl. Information! Please have a careful check yourself.

In conclusion, this is a fine study on the boundary between archaeology and paleoecology. Conclusions are of interest for a specific international audience. Relationships expressed in Fig. 4 are poor and this figure is superfluous.

Henry Hooghiemstra, Amsterdam, 25Jan2021

Appendix C

Molly Carney
Washington State University
Pullman, WA 99164
molly.carney@wsu.edu

Lianne Parkhouse, Leslie Brown, and Kevin Padian
Editors
Royal Society Open Science

Thursday, February 25, 2021

Dear Dr. Parkhouse,

Thank you for the opportunity to revise our paper on geophyte forager management entitled “Harvesting strategies as evidence for 4000 years of camas (*Camassia quamash*) management in the North American Columbia Plateau.” The suggestions offered by the four reviewers were immensely helpful in revising and editing the manuscript. Below, we detail how we have responded to each of these comments individually, indicating how we addressed each concern or problem and describing the changes we have made. For ease of reading, we have placed the reviewer’s comments in italics and our response to each comment below in plain text.

Our revised paper is now 6,842 words and includes 4 figures and 2 tables. The supplemental code and CSV files are archived online at <https://doi.org/10.5281/zenodo.4287583>.

Sincerely,

Molly Carney, Shannon Tushingham, Tara McLaughlin, and Jade d’Alpoim Guedes

Reviewer 1

This is an excellent manuscript, Molly et al. Well done! I have minor comments throughout and some suggested tweaks and references (see attached file 'RSOS-202213_Proof +NL.pdf').

Page 4 - 'palaeoethnobotanical' I think is more accurate?

Yes – thank you for catching this typo!

Page 4 - I would suggest both conscious and unconscious - ??

Great point. Edited MS to include conscious.

Page 4 - Make these phrases parallel tenses.

Edited sentence to same tense across examples.

Page 4 - Could add Lyons and Ritchie here re/ camas range extension on the Fraser River and its tributaries.

Added

Page 5 - hmmm, the most important plant food source?

Fair point. Added the qualifiers "among" and "plant food." For the Kalispel it was, but that does not mean everyone throughout the Northwest!

Page 6 - You could also potentially look at/reference parallel processes for balsamroot on the Canadian Plateau in Sandra Peacock's work.

Page 6 - Refs?

Extensively referenced above. Edited sentence to read: "Based on the previous literature review summarizing research on camas life cycles..."

Page 6 - I think you want some phrasing here to define this term. It's not defined, I don't think, until later in your methods.

Other than in the abstract, this is the first place in the manuscript where both terms are introduced. I think, to save space and word count, I'd like to leave the definitions here in the table. That way, it's easy to find as well.

Page 6 - Specify when this was.

Added dates (1985 and 1987 field seasons).

Page 7 - were grouped in?

You're right grouped in is redundant; I'd like to keep binned as it's common in statistics. Sentence now reads: "...dated features were binned into 500-year increments."

Page 8 - This section is well interpreted but looking at your data in Supp Table 1, you need a major caveat about sample sizes per time period--most data sets are <10 and 1 time period is only represented by one site. Did you consider larger time intervals? (like, early, mid, late?)

The problem with cultural chronology time periods on the Southern Plateau is that there are many versions. Every major watershed has its own chronology; Paul Solimano's big sticking point in many of his articles and talks is that someone needs to do the work in cross-referencing all of these. If we went with a larger southern Plateau chronology, the time spans are just too wide to catch any variation. We also are working on applying the methodology in this paper to other collections through the Pacific Northwest, with the intent of writing a follow-up paper. Our plan is to reuse those arbitrary 500 year intervals to circumvent some of these chronological challenges.

Page 9 - spell out

Thanks. Fixed.

Page 9 - And Gill's work can be referenced.

Reference 20 here refers to Gill (2014).

Page 9 - Again, Peacock's work would provide a useful comparison and further broaden the discussion.

Thanks for the suggestion. We added a few references to Peacock's work throughout to illustrate how our study complements her findings. We also have plans on applying this methodology to archived camas collections throughout the Northwest and writing a follow-up piece for an archaeologically-oriented audience, and will fully expand on her work when we do that.

Page 9 - I think you'll want to mention your sample sizes again as a qualifier and make sure not to overstep with your interpretations in light of this. You might also suggest research directions later in the paper.

Added sample size in to page 5. We also added a sentence on future directions in the TEK informs past and future section at the paper's conclusion.

Page 10 - Be careful with your wording throughout this section. At times it sounds a little climate deterministic, and I don't think that is your intent.

Thanks – yes, our main point is that we really don't see any correlations with climate and bulb size or harvesting strategy. We rewrote quite a bit of this section to emphasize this point further, based on your and another reviewer's suggestions.

Page 10 - Lyons and colleagues have an in press paper bearing on this question with respect to wapato that you might find useful.

Thanks for this suggestion – added the reference.

Page 11 - Several appropriate citations here, including Turner's new edited volume (2020), Reynolds and Dupres 2018, and others, Lepofsky's clam garden work, Deur's intertidal garden work (2015), etc.

Thanks for the suggestions – added.

Page 11 - As phrased, this sounds like the same thing, or at least, not significantly different; could you add a bit more descriptive phrasing to delineate and clarify, because the point is so important?

Rephrased to say: “We also demonstrate that for camas and other perennial geophyte plants, maturity at harvest is not synonymous with plant organ size, and we argue maturity has more utility in researching questions of cultivation, domestication, or management.”

Reviewer 2

This is an exciting study for Pacific Northwest paleoethnobotany and broader research into geophytes in the archaeological record. I am supportive of the overall interpretations and conclusions, with a question and minor suggestion to strengthen how you frame them within broader conversations about plant domestication:

In the section where you test your findings against Denham et al.'s (2020) suggested list of domestication syndrome traits in vegetative crops, I am puzzled by the interpretation that lacking evidence on this list suggests that "conscious or unconscious selection" is not involved (p.7, lines 58-60). Do you instead mean that you lack evidence for its role? And do you mean biological selection, as in factors/behaviors that led to "phenotypic and genotypic changes that are different from unmanaged populations" (p.3, lines 6-8)?

This paragraph is meant to show that we have mixed evidence for Denham et al.'s traits, and that we take the presence of some, but not all, traits as evidence for management and long-term human-camas relationships. We updated the paragraph here per your suggestions. Final sentence now reads: “Given this variation in phenotypic traits in archaeological and modern camas, we infer that the traits in Table 2 are the result of long-term management and human-camas entanglement rather than biological selection.” We also re-wrote parts of the paragraph to emphasize entanglement.

Though not necessarily evidence for biological selection, your study adds an important dimension to Denham et al.'s conclusions that phenotypic traits associated with early vegetative crop domestication is not ubiquitous. Maturity at harvest may not be a phenotypic trait associated with evolutionary adaptation, but it is (as you've pointed out) a physically recognizable variable for measuring the developmental stage of a geophyte in paleoethnobotanical evidence. By proxy, it can point to selective management practices. In a sense, your study is a strong case for how

whether camas were domesticated or not is beside the point - it does not diminish their entanglement with Indigenous communities in the Northwest. In light of your conclusions on p.9, I would suggest to lean more heavily into the significance of seeing "considerable continuity to human intervention into camas life cycles" (p.1, lines 38-39), when assessing your findings against Denham et al.'s list, and domestication studies in general.

That is the finding we are most interested in as well. We went through both the results and discussion section to emphasize selective harvesting as an alternative human-plant pathway.

Edited/added sentences throughout include:

Pg. 8 "Indeed, throughout our 4000 year study period we see little correspondence between paleoclimates and harvesting strategies as well as no evidence for change in bulb size or weight due to environmental conditions or other forms of biological selection."

Pg. 9 "These long-term entanglements or interactions should not be seen as incipient agriculture, but instead flexible relationships with plants that resulted in versatile yet sustainable subsistence practices."

Pg. 10 "We also demonstrate that for camas and other perennial geophyte plants, maturity at harvest is not synonymous with plant organ size, and we argue maturity has more utility in researching questions of cultivation, domestication, or management."

Again, this is incredible research - congratulations! I look forward to seeing it published.

Reviewer 3

In this manuscript, the authors present an interesting, well-planned, and generally well-executed study that shows evidence of changing strategies of camas management on the Columbia Plateau for the past 4000 years. The manuscript is for the most part well written, and I am generally convinced of the argument presented with regard to the camas/archy data and how collecting/management strategies seem to have changed through time.

With that said, I find the linkages presented between the camas data/archy data and the paleoclimate data a lot less compelling. I am unconvinced that the paleoclimatic data presented in in the study actually apply to this study area. I am not saying they don't, but this isn't clear from the manuscript as it reads now. In particular, there needs to be a better explanation of what the SST and d18O records are saying about past climate on the Columbia Plateau. Much of this might be accomplished by providing a more detailed caption for Figure 4. Once this is made clear, then the paleoclimatic data and the camas data need to be integrated in a more meaningful way so that the climatic changes discussed bear some relevance for the camas data. Right now it seems like the paleoclimatic information and the camas data are standing beside each other, but are not really talking to each other or showing any real linkages (see more specific comments below).

We rewrote several sentences and paragraphs of the Climate and camas consumption section given your comments here and specific comments below. For the Columbia Plateau, most discussions of plant food consumption, etc., are written as directly related to/totally dependent on climatic changes. Our main point is that we don't see any phenotypic changes in the bulbs as related to climate, nor do we see the harvesting strategies as climatically dependent. Our re-written sections throughout, including new intro and conclusion sentences for this section, are an attempt to emphasize this point. We also took care to make sure we discussed specific relationships (or lack of) between our datasets and the paleoclimate datasets. References to $\delta^{18}O$ and SST datasets were added.

We also updated the Fig. 4 caption. Indeed, one mistake of ours was to look to the paper title for the SST data in writing our caption; this led to some confusion across reviewers. Our SST dataset actually comes from Vancouver Island, which is not that far from our study site.

Lastly, I believe that the camas data would be more honestly represented in Figure 4 by bars than a time series because you binned the data into 500-year intervals. By having the data as single points connected by straight lines, it implies that there are increasing and decreasing trends between those data points throughout the time they span, and in reality this may or may not be true (but your data don't have the resolution needed to say one way or another). You should represent the camas data in figure 4 as bars (like a histogram) that span the correct time periods.

Thanks for this point. We edited the figure to include dot plots for the size ratio and number of leaves. Dot plots therefore help to illustrate sample size as well as more accurately showing the distribution of data across time.

My only other substantive comment is that you are very inconsistent with how you refer to time throughout the manuscript, in particular in the Results and Discussion sections. Cal yr BP, BP, cal BP, years, years ago. Go through and make sure these all match, especially when referring to calibrated radiocarbon years ("cal yr BP" is standard).

Ah yes thank you for pointing that out. We have gone through and made those changes throughout. Our apologies.

Specific comments to authors:

1. Summary

Page 2 Line 31: You say in the summary that you "bring together archaeological, paleoecological, and botanical data", but yet I don't really see any paleoecological data (for instance, pollen data) mentioned in the paper. I think perhaps you mean "paleoclimatological" data? Probably best to change the wording. I was expecting from the intro that you were going to discuss vegetation reconstructions for the late Holocene, but you didn't. This might be a good idea, however... (but likely beyond the scope of this paper).

Thanks for this point. We changed to paleoclimatological, and will look into incorporating the pollen data in future spin-offs.

Line 34: Need to make it clear at the start of the sentence, “Throughout the mid to late Holocene...” that you are talking about your results from this study. Further, I would characterize the past 4000 years as just the “late Holocene.”

Edited to read “In this region throughout the late Holocene...”

2. Introduction

Lines 53 and 54: Need a space between end of sentence and citation.

Thank you for catching this endnote typo – fixed.

Page 3 Lines 1-4: This is kind of a long and convoluted sentence. Can you shorten or break into multiple sentences to improve clarity?

Thank you for pointing this out. We broke that sentence into two separate sentences.

Line 4: “these alternative pathways” is a bit vague. Please clarify what you are talking about here.

Changing to human-plant relationships, and moving alternative pathways introduction below.

Line 17: Wouldn’t you say it is both “conscious and unconscious decisions and behaviors”? Think about including that word, too.

Yes absolutely. That is an oversight on our part, and another reviewer made the same point.

Line 28: Space needed before citation.

Thanks – fixed.

Line 29: Change “which” to “that.”

Thanks – fixed.

3. Human Relationships with Camas Life Cycles

Page 4 Line 26: I think it would be better to say “seasonally xeric moisture regimes.” Consider revising.

Added “seasonally” here. Thanks for the clarification.

Page 5 Line 10: Citation 36 is incomplete in the bibliography.

Thanks very much for catching that. Changed to in press.

Line 25: Comma needed after “knowledge.”

Thanks – fixed.

Line 26: Probably best to remove the comma and insert the word “and” before “replanting.”

Updated

4. Methods

Page 6 Line 25: I guess I am left wondering why you didn't just count them all. From what I can tell, it doesn't seem like that much work so there should be some sort of explanation why you didn't analyze all of the samples.

We included leaf scale data for all bulbs that were partially broken and which we did not have to dissect. For the whole (unbroken) bulbs, The Tribe wanted to only subsample a portion, saving some for potential future analyses. To respect those wishes, we agreed upon a 20% subsample. Dissecting or cross-cutting the charred bulbs typically resulted in broken leaflets and carbonized dust, so this 20% subsample was seemed the appropriate compromise between research and preservation.

Sentence now reads: “To preserve material for future analyses, we randomly subsampled...”

Lines 33-35: This looks strange to me. I would just write out the equation verbally in a sentence.

We opted to keep the equation, but are looking to the RSOS editors for guidance on this once. We defer to their authority on whether it's best to keep or change to a sentence.

5. Results

Lines 54-55: It took me a while to figure out whether your first sentence here is summarizing the literature or stating what you learned from your study. Please make it more clear that you are stating this based on your results. Also somewhat confusing is the clause “throughout the study period.” Consider revising.

Edited to read: “We found that people alternated between two camas harvesting strategies across the last 4000 years.”

Line 56: The statement “bulbs are different than both the mature bulb populations” is confusing. Are you referring to the control population? Please clarify.

Thank you for catching this – both should be control. Updated.

Line 60: I suggest inserting “i.e.,” before “stripping” in the parentheses. Same in other instances throughout manuscript.

Good point – updated throughout.

Page 7 Line 0: Need to make it clear here that you are talking about strategy (i.e., selective harvesting) in this sentence. Add language to indicate.

Clarified and added selective harvesting in here to indicate we discuss this strategy in this paragraph.

Lines 0-1: Rewrite to say “ 3499-3000 BP, 1999-500 BP, and 999 BP-present periods.” Be consistent.

Thanks – edited here, and did a scan for consistency throughout.

Line 5: Remove semi-colon and start new sentence at “We infer the most...”

Fixed.

Line 32: Remove the word “temporal.”

Deleted

Line 34: Comma needed before “respectively.”

Added

Line 35: Are you referring to the control population? Please clarify.

Added control here, and double checked control is specified throughout.

Line 36: End parenthesis missing. Need “BP” after “1999-500.” Delete “temporal.”

Thank you – added BP and deleted temporal.

Line 37: Add comma after “For the period,”.

Added.

Line 47-48: You say you “see no evidence for climatic influences on camas bulb size.” On what are you basing this conclusion? If you are going to state this here, then it needs some context.

We left it in for now in this location, since we wanted to keep results and discussion separate. But we wanted it there for the following climate and camas discussion, which was significantly revised per your suggestions. No other reviewer marked this sentence as out of place, but we are happy to rework if you would like to see that.

Table 2: Check weird superscript “7” in fourth column.

Thanks – fixed. That was weird.

6. Discussion

Page 8 Line 44-45: How do you know this is a period of “pronounced seasonality”? Need to include a brief description of where the data come from.

Added “...as determined by lake modelling analyses and $\delta^{18}O$ records” to the end of that reference, and there is the citation to Steinman et al.’s (2016) reconstructions.

Lines 48-50: This is a poorly written sentence and it makes it so the point you are trying to get across is not as strong as it could be. Rewrite to remove passive voice in both clauses.

Edited sentence: “We see two possible interpretations for the 4000-3500 BP “stripped” camas data: that past people had not established social institutions describing how these plants were harvested, or that they were not strictly following these cultural rules during a period of possible abundance.”

Lines 50-51: You need to make it clear how you know this. Is this based on the archaeological evidence, or is this an interpretation based on your study results? Language needs to be softened to make it seem less of a certainty and more of a likelihood that this happened based on what your data show.

Edited sentence to read: “Plateau archaeological summaries widely accept that practices of bulk-processing camas became fully routine after 3,500 cal BP.”

Line 53: “temporal period” is redundant. Please delete “temporal” here and in other places in the text.

Done

Line 53: Explain how you know that it was “residents and visitors” or revise language.

Modified sentence in 4.1 Archaeological Materials to emphasize that historically and further in the past, people traveled to the valley from all over the Northwest to harvest camas.

Lines 55-57: Again, this statement of “only” is a bit strong. Revise to indicate that this is what seems likely based on what your data show, but that you cannot know for certain.

Edited sentence to read: “Thus, by ~3500 cal BP it appears people were consciously and deliberately harvesting the bulbs of sexually mature plants...”

Lines 60- page 9 Line 1: You need to find a way to better weave together the paleoclimatic and camas data in this section. Instead of having separate sentences about climate and camas, you need to blend the ideas together to make a stronger statement. Also, if the “globally recognized cold period” (which honestly I don’t really think you have the data to show this since you SST record in from the NE Pacific), why mention it at all? Please rewrite this section.

Went through this entire section per your suggestions. Also edited caption for Fig 4 to reflect where SST data specifically comes from Vancouver Island.

Line 9: Be consistent with how you report years.

Thanks – edited throughout.

Line 10: Get rid of passive voice, “and there follows a 500-year period which primarily mature bulbs were collected.”

In making your suggested changes/rewrite of this section, we deleted this sentence. New sentence here reads: “The 2000-1500 BP data, however, indicate people primarily collected mature bulbs.”

Line 15: Cultural “changes”? If so, indicate.

Yes, thank you for pointing that out. Added.

Lines 15-26: Watch the tenses used in this paragraph. You switch back and forth several times.

Thanks – reviewed and made tweaks.

Lines 22-26: Again, the climatic description seems somewhat awkwardly thrown into this paragraph without providing any real context for how it applies to your data. What would these oscillations mean for camas or for people’s efforts to harvest it? If it is relevant, then you need to explain how. You also need to explain where you are getting the data for these oscillations (“as known from...”). The last sentence of this paragraph seems like a waste, and I would argue that it isn’t further climatic resolution that is needed, but a way to better integrate these data sets

given their vastly different sampling resolutions. Rewrite this section to better indicate how the observed climatic shifts either would or would not have influenced camas growth and management.

Double checked sources for “smaller decadal to centurial oscillations” and removed that sentence, primarily sticking with Steinman et al.’s $\delta^{18}O$ study. Moved comment on stabilized precipitation to first sentence and added citation to Steinman et al. 2016. Deleted last two sentences, and added a sentence to address your point about data set integration.

Line 28: Again, make it clear that you are basing what you are saying on your data.

This line is meant as a future studies preview. Added a sentence and citations; section now reads: “In future studies we plan on extending this methodology to archived archaeological collections of camas bulbs throughout the northwest, as well as looking into the phenology or seasonal timing of past camas harvests. We may also use these selective harvesting insights into human-camas relationships to model contemporary camas management and harvests, ensuring sustainable food options for Indigenous communities working to restore and reclaim autonomy over their health, well-being, and cultural heritage (62-65).”

Line 31: What do you mean by “archaeological evidence does not subside”? Please clarify.

Changed to “diminish.”

Line 34: On what are you basing your statement of “a subtle decrease in regional precipitation throughout the late Holocene”? Are you referring to SST data? If so, I don’t really see that in the data shown in Figure 4. The resolution of the record shown in panel B is much too coarse of a resolution to indicate that. If you are saying it’s shown by the $\delta^{18}O$ record, then you need to make that clear.

Edited sentence to read: “We suggest that the cultural institutions surrounding camas harvests ensured ongoing, sustainable harvests despite a subtle decrease in regional precipitation throughout the late Holocene [54, 62] and global cooling events such as the Little Ice Age [63].”

Page 10 Line 7: It seems strange to bring up “economic systems” here, considering that’s not really the focus of your study. Consider revising.

Edited sentence to read: “These long-term entanglements or interactions should not be seen as incipient agriculture, but instead flexible relationships with plants that resulted in versatile yet sustainable subsistence practices.”

Line 16: Change “which” to “that.”

Done

Line 23: Several of your results? Be clear and provide examples.

Edited sentence to read: “The identified “selective harvesting” practices over the past 1,000 years support ethnographic descriptions...”

Line 27: Revise so as not to repeat the use of the word “also” here, earlier in this paragraph, and at the start of the next paragraph.

Sentence now reads: “Furthermore, these results support Indigenous oral traditions...”

Lines 37-40: This statement seems like a stretch to me. I suggest deleting it or scaling it back.

The point of that sentence was to bring it full circle with our introduction. Edited to now read: “Such a perspective provides significant potential across research agendas spanning forager-plant relationships through time, the advent of cultivation practices, and in teasing out varied management/domestication pathways.”

Reviewer 4

This paper is tracking forager plant management practices. Research focus on harvesting practices by stripping versus targeting sexually mature camas bulbs, through the last 3500 years in the Pacific Northwest, USA. There is strong ethnohistoric evidence camas was a staple food in many Indigenous economies. Carbonized archaeological bulbs are examined for domestication syndrome traits.

-Fig. 1: ‘Kilometers’ should read ‘kilometer (km)’

-3/21: I guess the authors used old literature ? for the family name ‘Asparagaceae’ (Gould 1942). Following Mabberly (1993 and later editions) *Camassia* belongs to the Liliaceae.

[Mabberly, D.J., 1993, *The Plant-book*. Cambridge University Press, UK.]

To my knowledge, the genus was recently placed within the subfamily Agavoideae, in the family Asparagaceae (Chase et al., 2009). I also double checked theplantlist.org, which is what I’ve used in the past to check taxonomy and synonyms. MS is updated with the Chase citation, but happy to edit if I’m wrong.

-5/40: explain ANOVA and give a reference.

Added references for Mann-Whitney U / Wilcoxon test and Kruskal-Wallis rank sum test. Since ANOVA is very common across statistics, included in most introductory classes, we assume our

readers are familiar with this statistical test. We hope the added citations will direct readers with further questions to the appropriate sources.

-Fig. 2: I guess 'metric' suggests the units are 'cm'? Not clearly stated.

Added units in figure description.

-All over manuscript: while in the references years (e.g. 2014) are written without a comma, in the text years are written with/without a comma (e.g. 3,500 BP vs. 1766 BP). Better to eliminate this inconsistency, and, commas in numbers up to 10,000 are unnecessary and detract from readability.

Thanks very much – edited per your suggestions and another reviewer's.

-Table 2 is not introduced in the main text.

Thanks – added in text.

-Table 2: the acronym USO is not explained. To serve the reader, better not using acronyms here.

Thanks - changed to "underground storage organ."

-Figure 4: D18O = $\delta^{18}\text{O}$
panels B and C are lacking a unit.

Thanks – R doesn't work well with those symbols and we didn't catch it! Modified panel A in photoshop, and added units to the rest of the panels.

'Age BP' should read 'Age (yr BP)'

No one else mentioned this as an issue, so we did not change it. We did however, clarify units in the caption based on your suggestions and that of other reviewers. Hope that is an acceptable compromise.

-9/28 and elsewhere: in the text "cal BP" and "BP" is mixed up. Does this difference in units express a difference in precision? Please explain in the Method section the units of time.

This is our mistake. Went through the entire manuscript and edited to cal BP, as is the standard with the OxCal folks.

-The literature used (55, 56, 57) is outdated. I doubt if a comparison with sea surface

temperature from the Eastern Pacific makes sense as climate conditions may vary regionally.

The SST data was not that far geographically from our study site (off the coast of Vancouver Island) and was the closest SST paleoclimate dataset we could find. Our fault for not specifying that.

The “subtle decrease in regional precipitation”, the “global cooling events such as the Little Ice Age”, and panels ‘C’ and ‘D’ are not more than hand waiving arguments. Fig. 4 shows little information and can be deleted.

The main point of this climate and camas section is that we don't see any phenotypic changes in the bulbs as related to climate, nor do we see the harvesting strategies as climatically dependent. We went through this section carefully and re-wrote several sentences/re-structured several paragraphs to emphasize this point. We also updated panels C and D to dot plots to show sample size, and remove potentially erroneous linear relationships through time.

-I am missing a ‘Conclusions’ section clearly stating which incremental step forwards in understanding has been made. The current text in “Traditional ecological knowledge informs the past and the future” may be re-structured into a ‘Conclusions’ section.

We begin the paper by introducing traditional ecological knowledge in the abstract and introduction and wanted to circle back for the conclusion. The final section (6) is titled Discussion and Conclusions, with the subsection on traditional ecological knowledge as our way of bringing that topic back for consistency. We use the phrase “past informs the future” to lead into this final subsection which includes future avenues for research.

-References show some inconsistencies (e.g. PNAS in lines 11/44 vs. 13/39) (science = Science in 13/22)

-References: the abundant use of caps in titles is unnecessary (and inconsistently used).

Thank you! Realized we used the wrong Endnote citation style! Should be updated now.

I have not received Suppl. Information! Please have a careful check yourself.

In conclusion, this is a fine study on the boundary between archaeology and paleoecology. Conclusions are of interest for a specific international audience. Relationships expressed in Fig. 4 are poor and this figure is superfluous.

Thanks! We edited quite a bit of the climate and camas discussion to strengthen those connections. We also updated panels C and D in Fig. 4 to dot plots of size ratios and number of leaves per date. This shows sample size as well, and doesn't make the mistake of plotting linear relationships that may or may not exist. We hope this is an adequate compromise.